# Imaging Cu₂O nanocube hollowing in solution by quantitative in situ X-ray ptychography

Lukas Grote[1,2], Martin Seyrich [1,2], Ralph Döhrmann[2], Sani Y. Harouna-Mayer [1,3], Federica Mancini[1,4], Emilis Kaziukenas[1,5], Irene Fernandez-Cuesta [3,6], Cecilia A. Zito [1,7], Olga Vasylieva[1], Felix Wittwer [1,2], Michal Odstrčzil [8,10], Natnael Mogos[1], Mirko Landmann[2], Christian G. Schroer [1,2,9] & Dorota Koziej [1,3] ✉

Understanding morphological changes of nanoparticles in solution is essential to tailor the functionality of devices used in energy generation and storage. However, we lack experimental methods that can visualize these processes in solution, or in electrolyte, and provide three-dimensional information. Here, we show how X-ray ptychography enables in situ nano-imaging of the formation and hollowing of nanoparticles in solution at 155 °C. We simultaneously image the growth of about 100 nanocubes with a spatial resolution of 66 nm. The quantitative phase images give access to the third dimension, allowing to additionally study particle thickness. We reveal that the substrate hinders their out-of-plane growth, thus the nanocubes are in fact nanocuboids. Moreover, we observe that the reduction of Cu₂O to Cu triggers the hollowing of the nanocuboids. We critically assess the interaction of X-rays with the liquid sample. Our method enables detailed in-solution imaging for a wide range of reaction conditions.

Hollow nanoparticles with sizes in the range of several hundred nanometers comprise a material class with widespread application potential[1–6]. They are often utilized to form composite materials for high-performance electrodes in lithium-ion batteries[1,2], for (photo-) catalytic energy production[3,4], and as sensors[5,6]. Precise control over the structural and morphological evolution during the synthesis of these materials or during their operation in an electrolyte[7] is decisive to reach the desired functionality and high performance. In many cases, however, understanding and manipulating the course of the underlying morphological changes remains a major challenge. This is mainly due to the lack of experimental methods that can observe the dynamics of chemical processes in situ in a bulk solution under relevant conditions, such as high temperature and pressure. Visual insights into the formation dynamics of nanomaterials are thus rare.

Recently, efforts have been made to address this problem by exploiting in situ microscopy techniques. Liquid-cell transmission electron microscopy (TEM) plays an important role in this context[8–13], since it can provide spatial resolution down to the atomic scale[11]. In this

[1]University of Hamburg, Institute for Nanostructure and Solid-State Physics, Center for Hybrid Nanostructures, Luruper Chaussee 149, 22761 Hamburg, Germany. [2]Center for X-ray and Nano Science CXNS, Deutsches Elektronen-Synchrotron DESY, Notkestraße 85, 22607 Hamburg, Germany. [3]The Hamburg Centre for Ultrafast Imaging, Hamburg, Germany. [4]Institute of Science and Technology for Ceramics (ISTEC), National Research Council (CNR), Via Granarolo 64, 48018 Faenza (RA), Italy. [5]Department of Applied Mathematics and Theoretical Physics, University of Cambridge, Wilberforce Road, Cambridge CB3 0WA, UK. [6]Department of Physics, University of Hamburg, Luruper Chaussee 149, 22761 Hamburg, Germany. [7]São Paulo State University UNESP, Rua Cristóvão Colombo, 2265, 15054000 São José do Rio Preto, Brazil. [8]Paul Scherrer Institute, Forschungsstrasse 111, 5232 Villigen PSI, Switzerland. [9]Helmholtz Imaging Platform, Deutsches Elektronen-Synchrotron DESY, Notkestraße 85, 22607 Hamburg, Germany. [10]Present address: Carl Zeiss SMT, Carl-Zeiss-Straße 22, 73447 Oberkochen, Germany. ✉e-mail: dorota.koziej@physnet.uni-hamburg.de

way, the growth pathways of polyhedrally shaped Au nanoparticles[14] and the temperature-dependent shape evolution of Au–Pd core–shell nanorods[15] have been revealed. The high spatial resolution makes this method a powerful tool to study the growth of nanoparticles, however, the applicability of liquid-cell TEM to solution-based nanoparticle syntheses is limited, since it requires thin reactors with small volumes which can disturb the kinetics of the reaction and thus alter its pathway.

X-ray microscopy using synchrotron radiation overcomes these limitations[16–24]. Hard X-rays have the power to penetrate thick samples in extended environments[16], while at the same time allowing nano-imaging with a spatial resolution as high as 10 nm[25,26]. X-ray ptycho-graphy, which is a scanning coherent diffraction imaging technique, extends these benefits even further with the possibility of interpreting images quantitatively[27–31]. The method reconstructs the complex transmission function of the sample from a series of diffraction pat-terns using a phase retrieval algorithm. In contrast to scanning trans-mission microscopy, the spatial resolution achievable with X-ray ptychography is not limited by the beam size. The resulting images contain the local phase shift induced by the sample, which gives access to physical parameters such as the area density or, with some addi-tional assumptions, the density or thickness of the material.

Early in situ experiments demonstrated that lithium zirconate nanoparticles could be imaged in different gas atmospheres at ele-vated temperatures using X-ray ptychography[32], and the strain and deformation of a polymer–metal composite under pressure were quantitatively assessed in 3D by in situ X-ray ptychographic tomography[33]. Later, a phase transition during the melting of Sn–Bi alloy particles was recently visualized with a spatial resolution of 25 nm[18] and the annealing of Au nanoparticles was observed in dif-ferent gas atmospheres[19,21,22]. Additionally, X-ray ptychography allows the calculation of the wavefield at each point within the imaging setup. This enables the reconstruction of several individual objects, which are stacked along the beam direction, from a single two-dimensional (2D) scan. This procedure is referred to as multi-slicing[34,35]. In combination with the quantitative information contained in each image slice, we can obtain some three-dimensional (3D) insights into chemical processes at the nanoscale without the need for tomographic scanning.

In this study, we apply quantitative X-ray ptychography in solu-tion, visualizing the nucleation and growth of cuprous oxide ($Cu_2O$) nanocubes, and their subsequent transformation into hollow copper structures (Fig. 1)[36–38]. Using multi-slicing, we reconstruct separate 2D images of particles growing on the entrance and exit windows of the reactor, tracking the morphological evolution of individual nanocubes over time. We then access the third dimension by calculating the thickness of the particles from the quantitative phase images, allowing us to infer the full growth and transformation process in 3D. This way, we find different morphologies for particles that nucleate hetero-geneously on the reactor walls as compared to those nucleating homogeneously. At the end of the growth process, the nanocubes are reduced to metallic copper in a solid-state reaction[36]. At this point, we observe voids forming in the center of the particles and follow their expansion towards the surface, resulting in hollow nanocubes.

## Results

With the example of $Cu_2O$ nanocubes, we show how X-ray ptycho-graphy can be used for in situ nano-imaging of the shape evolution of nanomaterials in solution. To this end we developed a liquid reactor that provides the necessary mechanical and thermal stability for long-term imaging (Supplementary Note 1 and Supplementary Fig. 1). By simulating the thermal expansion of the reactor for small temperature fluctuations during operation, we find that thermal drifts of the reactor itself can be expected below 10 nm (Supplementary Fig. 2). While keeping the in-liquid beam path as short as 1 mm, the reactor contains a volume of 2 mL ensuring stable reaction kinetics.

The reaction of copper(II) acetylacetonate in benzyl alcohol is known to yield copper-based nanostructures with different morphologies[37,39], among those are $Cu_2O$ nanocubes which represent an intermediate product after the initial reduction of the precursor. Even though the subsequent reduction of the material to metallic Cu was observed[36], tracking the shape of the material at all reaction stages

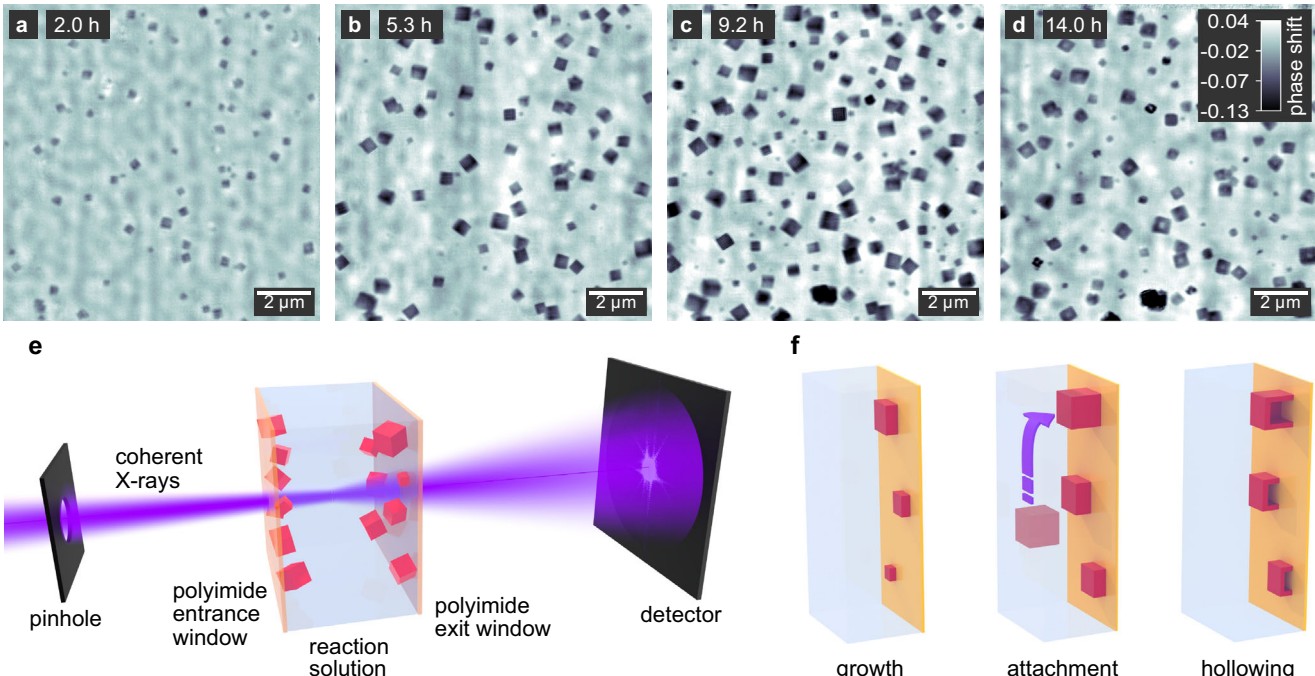

**Fig. 1 | Overview of the in situ imaging experiment. a–d** Ptychographic recon-structions of nanocubes on the entrance window of the in situ reactor. The gray scale indicates the phase shift in radian. **e** Schematic illustration of the experimental setup. A section through the reactor is shown for simplicity, including the entrance and exit windows that the nanoparticles grow on as well as the solution. **f** Illustration of the morphological evolution of the nanocubes as observed during in situ imaging.

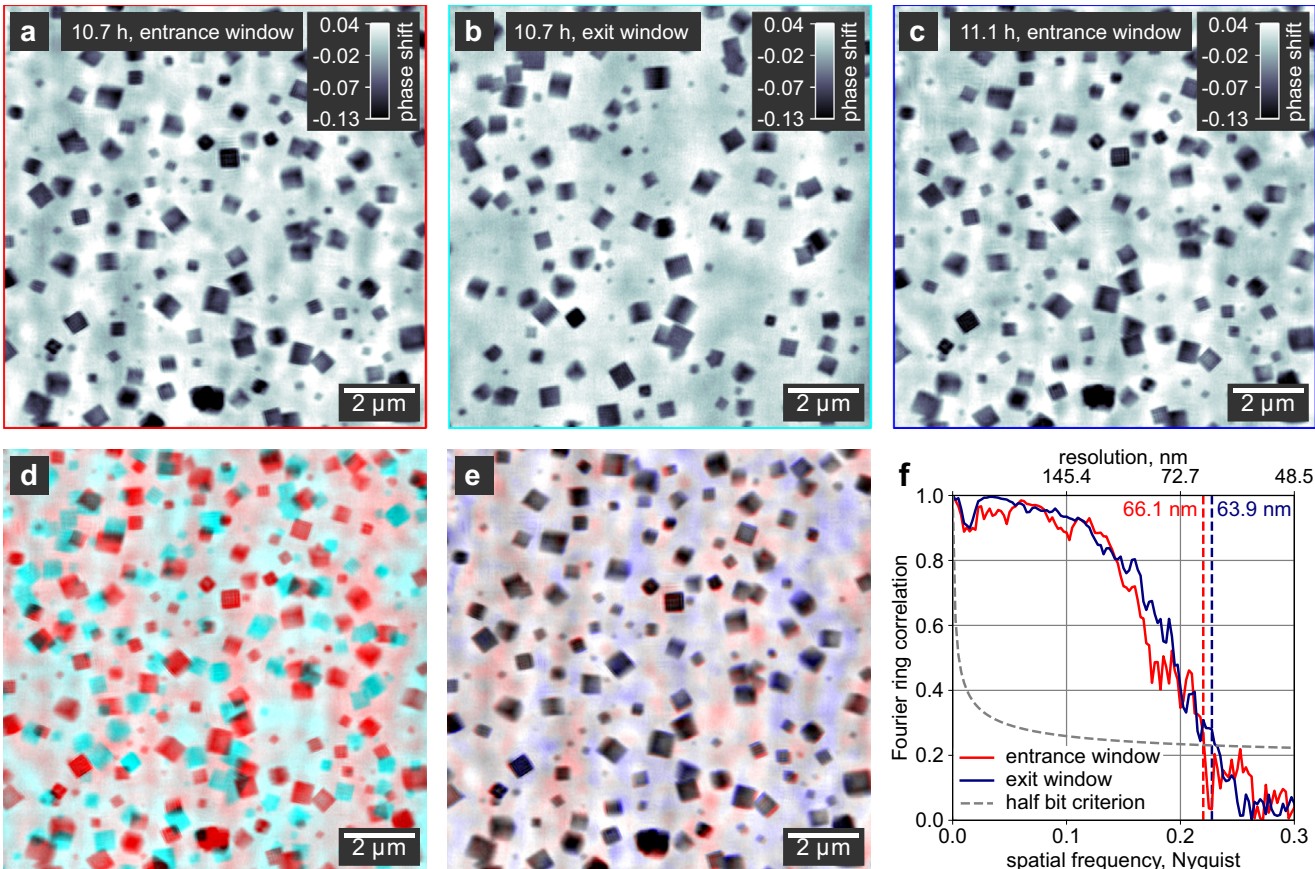

**Fig. 2 | Quality estimation of in situ images. a, b** Ptychographic images of particles growing on the entrance and exit windows, respectively, of the in situ reactor at the same reaction time. In **b**, the background fluctuations are varying on larger scales compared to **a** due to the larger illumination size on the exit window. **c** Ptychographic image of the entrance window at a later reaction time compared to (**a, b**). The gray scale indicates the phase shift of the image in radian. **d** False-colored overlay of images of the entrance (red) and exit (cyan) windows taken at the same reaction time. **e** False-colored overlay of aligned images of the entrance window taken at consecutive time steps of 10.7 h (red) and 11.1 h (blue). The background randomly fluctuates between images. **f** Fourier ring correlations (FRC) plotted for the entrance and exit window slices. We used the half-bit criterion[41] (dashed line) for resolution estimation to account for the loss in reconstruction quality induced by splitting the data set.

was never accessible in situ. To image the formation process of the nanocubes with X-ray ptychography, we fill the precursor solution into the in situ reactor and heat it to a reaction temperature of 155 °C. The nanocubes nucleate mostly on the walls of the reactor, which immobilizes them for sequential imaging (Fig. 1). We image the growth process at a time resolution of 21.6 min over 18 h (Supplementary Movie 1). Applying the multi-slice model[34,35] during the ptychographic reconstruction allows us to display the particles on the entrance and exit windows of the reactor separately. The particles on both windows are similar in size and morphology, we therefore show here only the nanocubes on the entrance window, unless otherwise noted. The respective images taken from the exit window are compiled in Supplementary Fig. 3.

## Quality of ptychographic images in solution

To determine the influence of the liquid reactor at high temperatures on the quality of the micrographs, we evaluate the spatial resolution, stability, and occurrence of background artifacts. We select ptychograms at intermediate reaction times to ensure a representative number of particles in the images. In addition, we evaluate the images taken on the entrance and exit windows of the reactor individually (Fig. 2a, b).

To estimate the spatial resolution, we use Fourier ring correlations[40] (FRC) displayed in Fig. 2f, which we obtain by splitting the diffraction patterns of a single ptychogram into two separate sets. We then reconstruct the sets individually and calculate the FRC of the two resulting images (cf. Supplementary Fig. 4). We apply the half-bit

criterion[41] compensating for the quality loss introduced by splitting the data set and find that the images of the entrance and exit windows have resolutions of 66 and 64 nm, respectively. The resolution is mainly limited by the reactor and the beam path of 1 mm in the hot liquid. Ex situ single-digit nanometer resolution images of $Cu_2O$ nanocubes can be achieved when measured in air[16,42].

Position stability is another important aspect of nano-imaging of chemical processes. To compensate for the thermal drifts of the reactor, we mechanically adjusted its position after each image. At the ptychographic reconstruction step, we applied automated position refinement[43] to correct for drifts of the reactor during the acquisition of a single image. Supplementary Fig. 5 shows a comparison of reconstructions with and without position refinement. After reconstruction, we performed additional image alignment (for details, see the "Methods" section). Figure 2e shows a false-colored overlay of two aligned images of the entrance window taken at consecutive time steps (Fig. 2a, c). Small remaining shifts of the particle positions are visible in the lower right area of the image, which does however not affect the information content or the later quantitative analysis of the images. This confirms that the substrate-bound nanoparticles in a heated solution can be imaged reproducibly and with high stability.

The main differences between the consecutive images are found in the background between the particles. We account the random background fluctuations to free-floating particles in solution or to convection of the hot solvent. Since the ptychographic model assumes a steady sample during the acquisition of one image, these non-static

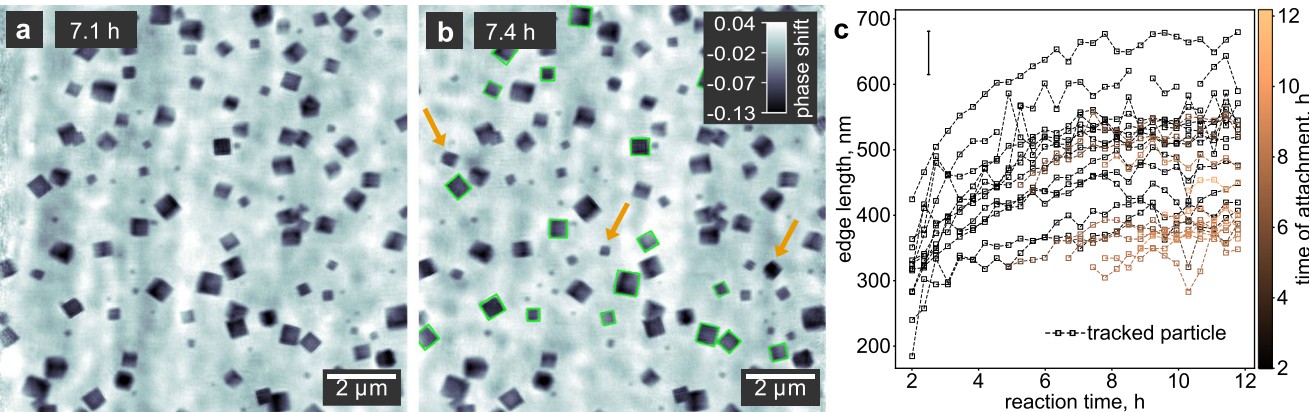

**Fig. 3 | Analysis of the growth process from a 2D perspective. a, b** In situ ptychographic images taken at 7.1 and 7.4 h, respectively. Particles marked with arrows appear in **b** but are not seen in **a**, indicating that they attached from the solution onto the reactor window. The gray scale indicates the phase shift of the images in radian. We used nanocubes highlighted with green rectangles for size estimation. **c** Evolution of the particle size for all nanocubes in **b** framed with a green rectangle. The color indicates the time when a particle attaches to the substrate. Independent of the attachment time, all particles show a similar growth behavior in the 2D projection. The error bar indicates the spatial resolution of the ptychograms of 66.1 nm.

portions of the sample cause inconsistencies in the ptychographic data set that result in a non-flat background. However, we can conclude that the inconsistencies do not significantly affect the reconstruction quality of the substrate-bound nanocubes. It is noteworthy that the predominant size of the background features is smaller in the reconstruction of the entrance window (Fig. 2a) compared to the exit window (Fig. 2b). The different feature sizes agree well with the diameter of the illuminating beam on the respective reactor windows (compare Supplementary Fig. 6).

The multi-slice approach allows us to reliably separate images of nanocubes growing on the entrance and exit windows of the reactor. Confirmation of proper slice separation is given by the false-colored overlay of the two windows at a single time step (Fig. 2d) showing that each particle appears in either of the two slices, but never in both. Due to a slow deformation of the polyimide windows, the distance between the slices gradually decreased during the reaction from initially 1000 μm to below 300 μm. Since an incorrect slice distance introduces artifacts to the reconstruction and lowers its resolution, we determined the distance several times during the reaction to ensure high image quality throughout the full-time series. For more details, see Supplementary Note 2 and Supplementary Fig. 7. When the distance between the windows reached 400 μm and thus approached the depth of field (DOF) of the imaging experiment of 280 μm, the slice separation started to be incomplete, with particles on the entrance window becoming dimly visible also in the reconstruction of the exit window (Supplementary Fig. 8). After about 17 h reaction time, the distance dropped below the DOF, and a multi-sliced reconstruction of the respective data sets was no longer possible. We thus treated the sample as a single slice at later reaction times, and reconstructions represent a superposition of the entrance and exit windows.

### Nucleation and growth of Cu₂O nanocubes in 2D

Particles are first visible after 2 h reaction time (Fig. 1a). Initially, with only a few small particles in the field of view, the quality of the reconstructed image is comparatively low, since the data set does not contain enough information for the iterative reconstruction algorithm to converge. After 5.3 h, the number of particles has increased, and they have grown to an average size of 450 nm. The image quality has also increased, and the cubic shape of the particles can be clearly distinguished (Fig. 1b). Since the ptychographic images represent projections of the electron density, we can recognize the orientation of the nanocubes when they reach their final size after 9.2 h (Fig. 1c). While some are oriented with one face parallel to the substrate, we can also see nanocubes facing towards the substrate with an edge or corner. There seems to be no preferential orientation. At a later reaction stage after 14 h, the particles develop a cubic central void (Fig. 1d), which suggests their reduction to metallic copper.

To determine the growth behavior in more detail, we track the size of the nanocubes throughout the in situ image series using image processing functions of the OpenCV library[44] (Fig. 3). Tracked particles are redrawn with a green rectangle in Fig. 3b, and the changing edge length is plotted for each particle in Fig. 3c. To ensure reliable results, we select only those nanocubes oriented with one face parallel to the substrate. A set of images with a smaller field of view is shown in Supplementary Fig. 9 for comparison and details on the image processing steps are given in Supplementary Figs. 10 and 11. The overall growth rate is fast in the beginning and successively slows down, resembling the pseudo-first-order kinetics that one would expect for this reaction of a single precursor with the solvent.

Most importantly, we observe that in addition to the nanocubes that nucleate on the substrate and are thus visible from the beginning of the reaction, new particles successively appear in the images. For example, within the time delay of 23 min between Fig. 3a and b, the three particles marked with yellow arrows appear. Since the new particles seemingly have similar sizes as those already present on the substrate, we can conclude that they must have nucleated homogeneously in solution and attached to the substrate at some point during their growth process. The fact that we see no particles detaching eliminates the possibility of nanocubes hopping into the field of view from a different area of the substrate. The free-floating particles are invisible to the in situ imaging since their motion in solution is faster than the acquisition time of a single image.

This interpretation is affirmed by inspecting the size evolution of newly attached particles only. In Fig. 3c, the time when a nanocube attaches to the substrate is encoded by the color of the respective data points. While the edge length of particles growing on the substrate since the beginning of the reaction is shown in black, the yellow color represents late attachment. It turns out that both the size and the growth rate of nanocubes attaching at intermediate and late reaction times integrate well with the overall behavior.

However, from a pure 2D perspective, we cannot differentiate between the morphological evolution of nanocubes attached to the windows of the in situ reactor and those dispersed in the solution. To this end, additional depth information is required.

### From 2D to 3D: Differentiating hollow nanocubes and -cuboids

As X-ray waves accumulate phase shift while propagating through matter, the thickness of an object along the beam path, i.e.

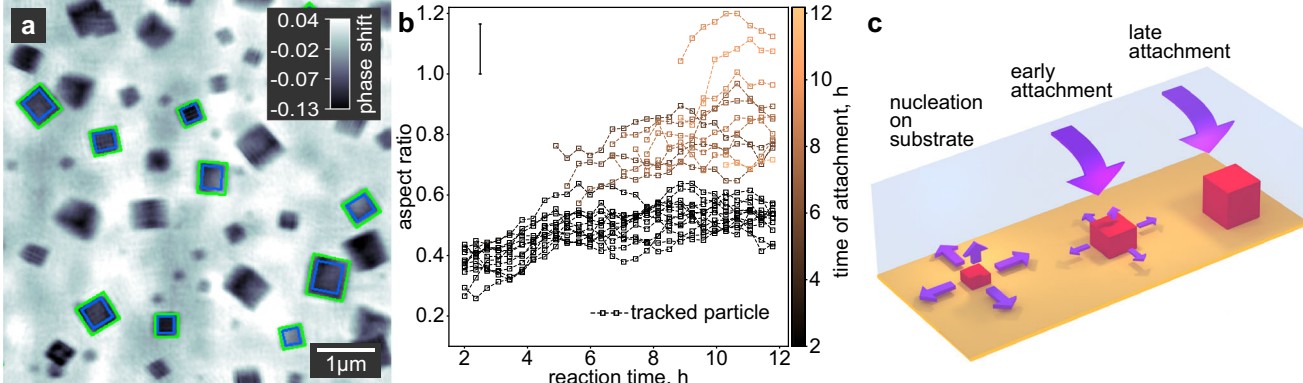

**Fig. 4 | Analysis of the growth process from a 3D perspective. a** In situ pty-chographic image highlighting the in-plane dimension of the nanocuboids by green rectangles. The out-of-plane thickness is calculated from the phase shift within the area highlighted by blue rectangles. The gray scale indicates the quantitative phase shift of the image in radian. **b** Evolution of the aspect ratio for all particles highlighted in Fig. 3b. The aspect ratio is calculated as the quotient of the in-plane dimension and the out-of-plane thickness of the nanocuboids. A moving average (MA) window of four time steps is applied. See Supplementary Fig. 12 for a plot without MA. The yellow color indicates the time when a particle attaches to the substrate. While the aspect ratio of individual particles does not change significantly with reaction time, it is strongly affected by the time when a particle attaches. The later the attachment takes place during the growth phase, the more cubic the respective particle becomes. This process is illustrated in **c**. For clarity, the error bar in **b** represents the mean error resulting from the error propagation of the in-plane particle size and the out-of-plane thickness of all tracked particles (for details, see Supplementary Note 3).

perpendicular to the image plane, can be calculated from a ptychographic image for known density and composition. This additional information allows us to put together a 3D model of the sample. Given the X-ray wavelength and the electron density of the material, we calculate the thickness of each tracked particle from its phase shift by taking the average of the pixel values across individual particles in the quantitative ptychograms. To account for the finite resolution, we exclude pixels close to the edges of a particle, as it is exemplarily shown for a single time step in Fig. 4a. Since the sizes of the nanocubes vary, we use the aspect ratio rather than the absolute thickness to describe the 3D shape of each particle. The aspect ratio is calculated as the quotient of the edge length in the image plane and the out-of-plane thickness. The fluctuating background induces an error for the calculation of the aspect ratio at a given time, but it does not change the observed general trends in the evolution of particle shape (see Supplementary Note 3).

In Fig. 4b, we see that the aspect ratio of individual nanocubes remains nearly constant, however, a systematic trend is visible with regards to the time a particle attaches to the substrate, which is highlighted by color. Those particles that attach when growth has almost finished at late reaction times (yellow) are cubic. However, the longer a particle grows on the substrate, the more it deviates from an ideal cubic shape with the edge length perpendicular to the substrate being much shorter. Particles that are visible from the beginning of the reaction and thus presumably nucleate on the substrate (black data points in Fig. 4b) are cuboids with an aspect ratio of about 0.5. For the growth mechanism, this means that the particles grow from only one face in the direction perpendicular to the substrate, while they grow parallel to the substrate equally from all four faces, as illustrated in Fig. 4c.

We account for the apparent increase in the aspect ratio from about 0.3 to 0.5 during the first 5 h of reaction time in Fig. 4b to a systematic error in the calculation of the thickness for these particles. Since the nanocuboids are small at the beginning of the reaction, in combination with the limited resolution, the fading of the phase shift signal at the edges of the nanocuboids leads to an underestimation of the thickness for these early reaction times.

Once we have derived the growth mechanism in 3D from the quantitative phase images, we can begin to understand the hollowing process of the nanocuboids bound to the substrate (Fig. 5). After ca. 11.8 h, the particles do not grow significantly larger (compare Fig. 3c) and we can assume that most precursor has been consumed. At this point, cubic voids start to form in the center of the particles and grow towards the surface over the next 3.2 h (Fig. 5a–f). We confirm that the hollowing process is not due to the X-ray exposure by taking a single ptychographic image at the end of the reaction at a different part of the reactor window, ca. 2 mm away from the previously exposed region (Supplementary Fig. 13). This image also shows hollow particles.

Even though the reduction of the $Cu_2O$ nanocuboids to metallic Cu was known[36,39], previous in situ measurements never observed hollow particles due to their lack of depth information. X-ray ptychography can now enable us to also probe the internal morphology of the material.

When $Cu_2O$ is reduced in the solid state to Cu, there must be an outward flow of oxygen species caused by the reaction with benzyl alcohol taking place at the particle surface. Recently, Kirkendall hollowing was found during the fast extraction of phosphorous from $PdP_2$ nanoparticles[45]. In $Cu_2O$, however, the diffusion coefficient of $Cu^+$ cations is much higher than that of $O^{2-}$ anions[46]. This difference in diffusion rates would only cause void formation during oxidation of Cu to $Cu_2O$, but we observe the hollowing process in the course of reduction. Due to the slow void formation in our nanocuboids during 3 h, it is unlikely that the extraction of oxygen species from the nanocuboids has a significant effect on the diffusion rates. Thus, the diffusion of ionic species and therefore the Kirkendall effect[47] are unlikely the cause of void formation in the nanocuboids. Instead, interactions with the solvent and the substrate at the surface of the nanocuboids protect their outer shape during reduction. The binding of the $Cu_2O$ nanocuboids to the polyimide substrate indicates a strong interaction between the two materials, and furthermore, strong adsorption of benzyl alcohol at the surface of $Cu_2O$ was observed in a previous in situ spectroscopic study[39]. Due to these surface interactions, the mass reduction upon depletion of oxygen from the crystal lattice and, more importantly, the increase in density from 6.0 g/cm³ ($Cu_2O$) to 8.92 g/cm³ (Cu)[48] are compensated by a central void instead of shrinking of the outer particle dimension[49,50].

A similar effect was observed during the reduction of CoO particles to form hollow Co shells in the presence of oleylamine[51], as well as for the reduction of β-FeOOH to hollow $Fe_3O_4$ nanorods with strongly bound capping agents[52]. Since no more nanocuboids are attached to the reactor windows after the beginning of the void formation, we cannot conclude if dispersed particles undergo the same shape transformation.

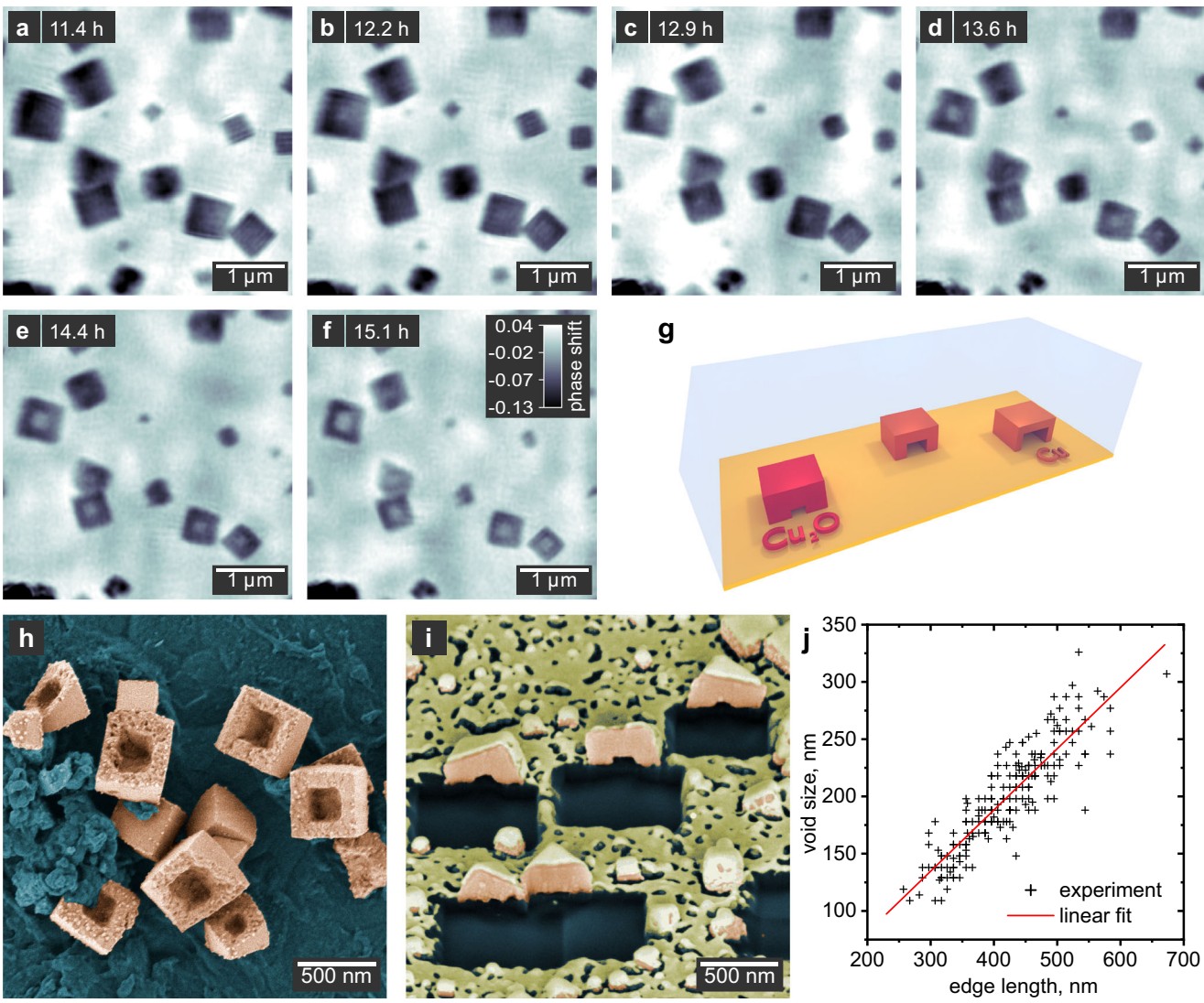

**Fig. 5 | Hollowing process of the nanocuboids. a–f** Ptychographic images of the hollowing process between 11.4 and 15.1 h reaction time. The gray scale indicates the phase shift of the images in radian. **g** Schematic illustration of the void formation. **h** False-colored SEM image of bowl-shaped hollow nanocuboids (orange) taken from the same sample used for the in situ imaging in **a–f**. **i** SEM image of hollow nanocuboids cut in half using a FIB. These particles were prepared in the in situ reactor but without exposure to X-rays. The sample was sputtered with a thin Au layer (yellow) to minimize charging effects. In the areas cut with the FIB, the substrate can be seen (blue). **j** Scatter plot of the void size versus the outer particle dimension obtained from SEM images of the same sample used for the in situ imaging in **a–f**. The fit indicates a systematic size ratio of 53%.

To further verify our interpretation, we compare the edge length of the cubic voids to the outer edge length of the particles at the end of the reaction (Fig. 5j and Supplementary Fig. 14). Independent of the absolute sizes, we find a ratio of approximately 53%, which underpins the void formation to originate from a specific chemical transformation. However, assuming full conversion from $Cu_2O$ to Cu, we would expect a size ratio of 73.8%, indicating the presence of some residual oxide in the final hollow structures.

To verify the role of the substrate in the hollowing process of nanocuboids, we examined the particles grown on the reactor window during the in situ measurements in the SEM. The false-colored image in Fig. 5h shows that the particles in fact have the shape of rectangular bowls. Due to their flatness, it is reasonable to assume that the hollowing process starts at or close to the particle-substrate interface, and then carves into the particle in the direction away from the substrate. To further test this hypothesis, we use a focused ion beam (FIB) on a fresh sample not exposed to X-rays and cut some nanocuboids in half that are still bound to the substrate (Fig. 5i). It is clear that the void is located in between the bowl-shaped copper particles (orange) and the

substrate (blue). Thus, the particles in Fig. 5h must have detached from the substrate and turned around before the image was taken. The void formation process is schematically illustrated in Fig. 5g.

**Photon energy dependent radiation damage in X-ray imaging**

We have shown that hard X-ray ptychography can reveal the shape evolution of hollow copper nanocuboids with a high level of detail. However, when imaging nanostructures in chemically reactive environments, alterations of the sample induced by the probe beam are a common obstacle in both TEM[12,53] and X-ray microscopy[54,55]. In our case, the partial absorption of the X-ray beam within the reaction medium leads to a local energy deposition that can cause changes in the reaction pathway. We found that the photon energy has a strong influence on the characteristic of the radiation damage. At an energy of 8.98 keV, which is just above the K edge of copper, the X-ray beam triggered the nucleation and attachment of nanoparticles at the reactor windows. As shown in Supplementary Fig. 15, almost no particles are found outside the illuminated area after 18 h reaction time. Also, in a different experiment under the same conditions, micrometer-sized

spherical structures formed under illumination instead of the expected nanocuboids (Supplementary Fig. 16). This indicates that the comparably high absorption at 8.98 keV fully reduces the Cu(acac)$_2$ precursor already in solution, causing a deviation from the usual reaction pathway.

Therefore, we use the higher photon energy of 15.25 keV for all in situ images, reducing the influence of the X-ray beam on the reaction. At this energy, the attenuation coefficient of Cu$_2$O is about 4 times lower than at 8.98 keV, significantly reducing the amount of energy deposited by the beam[56]. Although the field of view was under repeated exposure for continuous imaging, we neither noticed any influence of the beam on the nucleation or attachment of particles nor was the shape or size of the nanocuboids affected during growth. We showed that the void formation coming along with the solid-state reduction is also not a beam effect. However, after the void has formed, the particles shrink and finally vanish, starting at 16 h (Supplementary Fig. 17a–c). This suggests that metallic copper is less stable under irradiation due to its increased density and lower chemical stability compared to Cu$_2$O. The metallic particles undergo side reactions with the organic solvent forming soluble copper species. At 20.5 h reaction time, we extended the field of view around the region that was continuously exposed before (Fig. 6a). The disintegration of metallic copper particles is localized to the exposed region. The SEM image of a region in close vicinity to the exposed area (Fig. 6b) shows nanocuboids with truncated edges. It is plausible that the ablation sets in at the edges where the higher surface energy facilitates side reactions.

While the absorption of X-rays in matter is significantly weaker compared to that of electrons, the high flux and long acquisition time currently required for scanning-based X-ray microscopy diminish this benefit. Still, by choosing high photon energy, we prevented radiation damage during all reaction stages until the formation of the final hollow structures and could thereby visualize the full undisturbed shape evolution.

## Discussion

Recent in situ studies using X-ray spectroscopic and scattering techniques revealed that the nucleation of nanoparticles is followed by complex structural and chemical changes in solution[57–60], including transformations of distinct primary particles[61,62], assembly[63,64], and hollowing processes[46,65]. These findings highlight that many pathways exist that lead to complex, hierarchical materials, besides the classical LaMer model of nucleation and growth[66]. However, since X-ray scattering and spectroscopic methods provide information averaged over a given reaction volume, only in situ microscopy gives proof of these pathways. We demonstrate how in situ imaging in solution with hard X-ray ptychography enables direct visualization of the complex transformations of shape and size at the nanoscale. The method can be applied to a wide range of materials and reaction conditions, complementing the capabilities of liquid-cell TEM. Such rare visual insights into structural changes in solution are important to deepen our understanding of the origins of nanomaterials morphology, which is a key factor for the future design of highly active catalysts and functional devices.

In summary, we revealed the full morphological growth of Cu$_2$O nanocuboids and their subsequent transformation into hollow Cu structures by directly visualizing them in situ inside our chemical reactor with an in-liquid beam path of 1 mm. Making full use of the quantitative phase images, we found that the 3D shape of the nanocuboids is strongly influenced by their interaction with the walls of the reactor. When attached to the substrate, the out-of-plane growth is hindered, and the particles grow into flat cuboids with an aspect ratio of 0.5. Particles nucleating in solution and attaching to the substrate at a later growth stage take a more cubic profile. The hollowing process, which we assign to the solid-state reduction of Cu$_2$O to Cu, originates close to the particle–substrate interface. The voids grow towards those

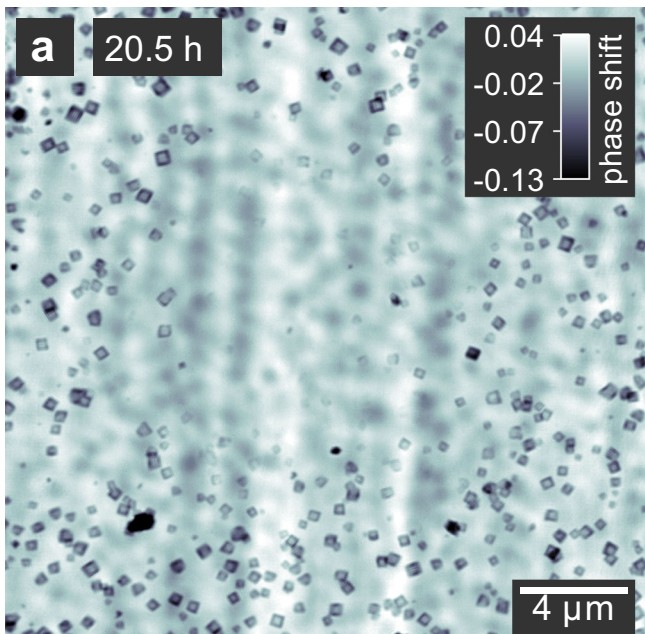

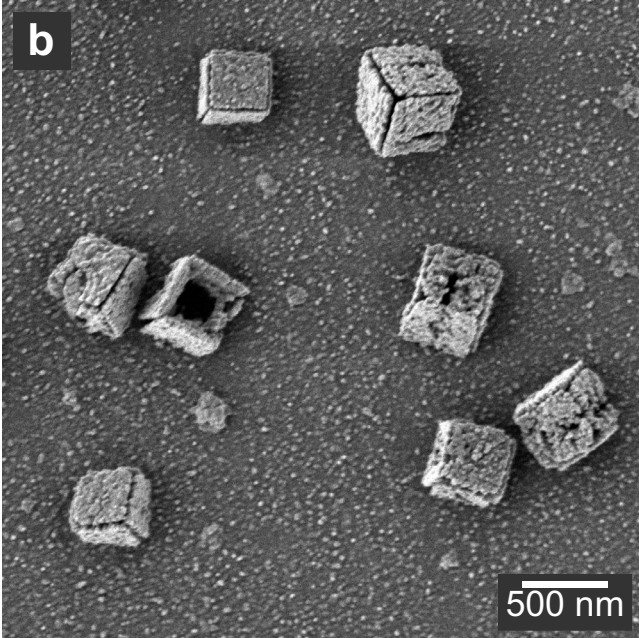

**Fig. 6 | Beam-induced damage to the nanocuboids. a** Ptychographic image taken after 20.5 h with an extended field of view around the region that was continuously exposed for in situ imaging. The beam-induced disintegration of particles is visible in the exposed region in the middle. The image shows an overlay of particles on both reactor windows since multi-slicing could not be applied due to the decreasing distance between the windows. The gray scale indicates the phase shift of the image in radian. **b** SEM image of particles from the same sample as in **a**, taken close to the exposed region. The nanocuboids show the onset of disintegration at the edges.

faces that are in contact with the solution, i.e., where the reducing reaction with the solvent takes place, and leave the particles with a bowl-like shape. Finally, we identified the influence of the beam on the reaction at different photon energies and minimized the beam damage by choosing a high photon energy with correspondingly low energy deposition. Under these conditions, only the hollow metallic structures that denote the end of the reaction are sensitive to the beam, but no deviations are observed along the reaction path.

According to our observation of reduced beam damage at the higher photon energy of 15.25 keV, we expect that beam damage will be further reduced with the emergence of fourth-generation synchrotron sources providing highly coherent X-rays at even higher energies than 15.25 keV. At the same time, the high coherent flux of these new sources can significantly increase imaging speed and spatial resolution. Already now, the high level of microscopic detail obtained within our in situ X-ray microscopy experiment encourages the application of this approach to explore shape evolution and transformation processes in related fields, such as the formation of hollow metal oxide nanospheres composited with graphite or sulfur for lithium-ion batteries[1,2], or with noble metal particles for (photo-) catalytic energy production[3,4]. Also, biomineralization processes[67–69] or syntheses of bio-inspired nanomaterials[70,71] may be worth visualizing.

## Methods

### Synthetic procedure

**Chemicals.** Benzyl alcohol (>99.8%, anhydrous) and Cu(acac)$_2$ (99.99%) were purchased from Sigma-Aldrich, and ethanol (absolute) for washing from VWR. All chemicals were used without further purification.

**Synthesis.** 1.72 mL of benzyl alcohol was added to 0.03 mmol Cu(acac)$_2$ and stirred for 10 min at room temperature in a glove box under an argon atmosphere. The blue solution was transferred to the reaction container of the in situ reactor, sealed, and heated at a rate of 4.5 °C/min to 155 °C. The reaction was stabilized at this temperature for a maximum of 30 h for in situ imaging, and for FIB experiments the reactor was cooled to room temperature after 14.5 h at a cooling rate of 5.5 °C/min. All mentioned reaction times are relative to the point when we start heating. For SEM and FIB, the polyimide windows of the reactor were rinsed thoroughly with ethanol and dried at 60 °C overnight.

### Ex situ characterization

**SEM.** Scanning electron microscopy images were taken with a Regulus 8220 (Hitachi High Technologies Corp., Japan) at an acceleration voltage of 3.6 kV and under the application of a deceleration voltage of 3.5 kV, resulting in a maximum electron energy at the sample surface of 0.1 keV. Secondary and backscattered electron signals were mixed.

**FIB.** Focused ion beam milling and subsequent SEM imaging were conducted with a Crossbeam 550L (Carl Zeiss AG, Germany) using a Ga ion beam at 30 kV and 50 pA, and an electron probe at 5 kV.

### In situ reactor

Detailed information about the in situ reactor including a schematic is given in Supplementary Note 1. The container for the reaction solution is made of PTFE and two polyimide windows with a thickness of 125 μm. The container has a volume of 2 mL and, when assembled, the beam path through the liquid is 1 mm. The container is clamped into a heatable body made of invar and titanium for low thermal expansion. A PID controller is used to keep the reactor at the reaction temperature with a stability of ±0.05 °C.

### X-ray ptychography

**Beamline setup and data acquisition.** Measurements were performed at the Ptychographic Nanoanalytical Microscope PtyNAMi[72] at the nanoprobe endstation of beamline P06[73] at PETRA III at DESY in Hamburg, Germany. A coherent X-ray beam at a photon energy of 15.25 keV was focused using stacks of nanofocusing refractive lenses. An Eiger X 4M (Dectris AG, Baden-Daettwil, Switzerland) photon counting area detector with a pixel size of 75 × 75 μm² was placed at the end of an evacuated flight tube 3.435 m behind the sample. For a ptychographic image within a time series, the in situ reactor was raster scanned perpendicular to the beam within a field of view of 12 × 12 μm² with a step size of 300 nm. Coherent far-field diffraction patterns were measured with an exposure time of 0.5 s. Thermal drifts of the reactor were compensated by adjusting its position between image acquisitions. Each scan took 21.6 min. For the larger image in Fig. 6a, a field of view of 24 × 24 μm² was scanned.

For the images shown in Supplementary Figs. 15 and 16, measurements were taken at the cSAXS beamline at the SLS at PSI, Villigen, Switzerland. A photon energy of 8.98 keV was used and the beam was focused with a Fresnel zone plate in combination with an order-sorting aperture. At the end of an evacuated flight tube, a Dectris Pilatus 2M detector with a pixel size of 172 × 172 μm² was placed 7.35 m behind the sample. The field of view was 10 × 10 μm² for an in situ image series, and 30 × 30 μm² for the larger image shown in Supplementary Fig. 15.

**Image reconstruction.** For reconstruction, the diffraction patterns were cropped to 256 × 256 px, resulting in a pixel size in the reconstructed images of 14.54 and 23.06 nm for the measurements at P06 and cSAXS beamlines, respectively. The image reconstruction of data taken at beamline P06 was carried out using the extended ptychographic iterative engine (ePIE)[74]. This technique is the subject of patents owned by Phase Focus Ltd. and the University of Sheffield[75–77]. For the data from cSAXS, the difference map algorithm in combination with a maximum-likelihood approach was used[78,79]. For data taken at P06, the object was initialized to be non-absorbing and non-phase-shifting, and a Gaussian probe with a FWHM of 1 μm and a phase curvature of 2.2 mm was used as the initial probe. For images containing few particles early in the time series before 3.5 h, the illumination was initialized with the reconstructed illumination of the image after 3.83 h. For data measured at the cSAXS beamline, a reconstructed illumination from an ex-situ sample was always used as an initial probe. Multi-slice reconstructions were carried out for the P06 data. Here, the object slice corresponding to the entrance window was initialized first, and the exit window slice was activated after 20 iterations. From iteration 250, position refinement was performed every 100 iterations with a maximum displacement of 44 nm[43]. Reconstructions were run for 1000 iterations. The distance between the slices was initially set to 1 mm and gradually reduced, adapting to the shrinking distance between the windows of the in situ reactor (for details, see Supplementary Note 2) The Fresnel nearfield propagator[80] was used for wave propagation between slices.

### Image analysis

**Alignment.** To compensate for thermal drifts of the in situ reactor, in addition to the mechanical compensation described above, we aligned the image series using an implementation of the Scale-invariant Feature Transform (SIFT) method[81] within the image analysis software ImageJ[82] (version 1.53k).

**Fourier ring correlation.** To calculate FRCs[40,41], the diffraction patterns of the image were split into two data sets with equidistant scan points and reconstructed individually, with the refined positions from the full data set. The FRC was calculated from the resulting two images. We used the half-bit criterion for resolution estimation in order to account for the lower resolution introduced by splitting the data set.

**Depth of field.** In the multi-slice model, the DOF can be considered as the maximum thickness that an object may have to be imaged within one slice at a given lateral spatial resolution $\Delta d$ and wavelength $\lambda$. We calculated the DOF using the empirical relation[34] given by

$$\mathrm{DOF} = 5.2 \frac{(\Delta d)^2}{\lambda} \tag{1}$$

where $\Delta d$ is the lateral resolution and $\lambda$ is the X-ray wavelength. Here, we calculated a DOF of 280 µm.

**Quantitative image analysis.** To track and measure the dimension of cubic particles within the image plane, several image processing steps were applied using Python and the PyOpenCV library[44] (version 4.5.1.48). First, images were converted to binary (black and white) by applying adaptive thresholding with a Gaussian kernel with a size of $\frac{1}{8}$ of the image dimension. Closing (dilation followed by erosion) with a kernel of size $(2 \times 2)$ was used to reduce image noise, and contours (borders between black and white areas) were identified. The smallest rectangle containing all points of a closed contour was taken as the outer dimension of the respective particle. Only particles visibly oriented with one face parallel to the substrate and with an in-plane aspect ratio between 0.7 and 1.3 were tracked. An example image after each of these steps is shown in Supplementary Fig. 10.

For calculating the thickness of a particle from a quantitative phase image, first, the average background phase shift $\phi_{bkg}$ of the image was calculated by applying a threshold of $-0.02$ rad to separate particles and background (Supplementary Fig. 11), and by then averaging the background. Only for particles with one face visibly oriented parallel to the substrate, the average phase shift $\phi_{particle}$ was calculated within an area covering 70% of the full particle dimension to account for the not perfectly sharp edges (blue rectangles in Fig. 4a). The thickness $d$ was then calculated using Eq. (2)

$$d = \frac{-\left(\phi_{particle} - \phi_{bkg}\right)\lambda}{2\pi\delta} \, , \qquad (2)$$

where $\lambda$ is the X-ray wavelength and $1 - \delta$ is the real part of the complex refractive index. We used the difference of the refractive index decrement of $Cu_2O$ and the surrounding benzyl alcohol, resulting in $\delta = 3.98 \times 10^{-6}$ [83].

**Reporting summary**

Further information on research design is available in the Nature Research Reporting Summary linked to this article.

## Data availability

The X-ray ptychography and SEM data sets generated and analyzed during the current study are available in the zenodo repository, https://doi.org/10.5281/zenodo.6675817[84].

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

## Acknowledgements
This work was supported by the Bundesministerium für Bildung und Forschung (BMBF) within the Röntgen-Ångström-Cluster via the project 05K2020—2019-06104 XStereoVision (grant no. 05K20GUA, L.G., D.K., C.G.S.), by the European Research Council within the project LINCHPIN (grant no. 818941, C.A.Z., D.K.), and by the Cluster of Excellence "CUI: Advanced Imaging of Matter" of the Deutsche Forschungsgemeinschaft (DFG)—(EXC 2056—project ID 390715994, S.Y.H.-M., I.F.-C., D.K.). We thank Dr. Maik Kahnt and Stephan Botta for supporting the X-ray measurements and Prof. Robert H. Blick for providing access to the FIB-SEM device. We also thank Prof. Marcin Sikora and Krzysztof Pitala for EM imaging. We acknowledge DESY (Hamburg, Germany), a member of the Helmholtz Association HGF, for the provision of experimental facilities. Parts of this research were carried out at PETRA III and at the DESY Nano-Lab. We would like to thank Dr. Gerald Falkenberg and Dr. Jan Garrevoet for assistance in using beamline P06, and Dr. Thomas Keller, Satishkumar Kulkarni and Arno Jeromin for laboratory support. We acknowledge the Paul Scherrer Institut, Villigen, Switzerland for provision of synchrotron radiation beamtime at the cSAXS beamline of the SLS and would like to thank Dr. Andreas Menzel, Dr. Ana Diaz and Xavier Donath for assistance.

## Author contributions
D.K., C.G.S., and L.G. conceived the project. L.G. and R.D. designed the in situ reactor. M.L. conducted the thermal simulations of the in situ reactor. L.G., N.M., S.Y.H.-M., and F.M. prepared the synthetic procedure. L.G., M.S., S.Y.H.-M., C.A.Z., O.V., F.W., and M.O. conducted the X-ray ptychography experiments. L.G., E.K., F.M., and I.F.-C. conducted the FIB milling and SEM imaging, and analyzed the images. L.G. and M.S. reconstructed the ptychographic images. L.G. conducted the quantitative image analysis. All authors contributed to the data interpretation and the preparation of the manuscript.

## Funding

## Competing interests
The authors declare the following competing interests: The ptychographic imaging technique used in this paper is the subject of awarded patents owned by Phase Focus Ltd., which is a spin-out company of the University of Sheffield. Phase Focus Ltd. does not restrict the use of the technique for academic research. None of the authors is affiliated to the company. These competing interests exist for all the authors.
