## [Peer Review File · Nature Communications]

This manuscript by Grote *et al.* presents the use of X-ray ptychography to study the growth of Cu₂O nanocubes and the formation of hollow structure during the reduction of Cu₂O to Cu. In general, this is an interesting work which shows a powerful demonstration of the utility of X-ray ptychography in *in-situ* characterization of materials growth & reaction. Here are some of my comments that could improve the work:

1. The use of “in 3D” in the title is misleading. Even though multi-slice ptychography was used in this paper to distinguish the entrance and exit planes that were separated by more than 280 microns, only 2D slices (e.g. the entrance window) which are the “projections of the electron density” as stated in line 165 were used for the data analysis.
2. Authors should consider including some early papers of in-situ X-ray ptychography in the introduction. For example:
Høydalsvik *et al.* <https://doi.org/10.1063/1.4884598>
Fløystad *et al.* [https://DOI: 10.1002/adem.201400443](https://DOI:10.1002/adem.201400443)
Strelnikova *et al.* <https://DOI:10.1038/s41598-017-13175-9>
3. In line 87-88, it is not quite clear what the “single-digit nanometer range” is?
4. In contrast to “Irrespective of small shifts” and “position of the nanocubes is stable”, obvious shifts can be found in Figure 2e between two consecutive scans acquired in 10.7 h and 11.1 h. For example, the blue and red squares are unmatched especially upper left of the scale bar. (1) Was this shift due to the reactor’s thermal drift? (2) A striking information provided in the manuscript was that the two polyimide windows were deformed during the reaction so that the cell thickness decreased from 1000 μm to below 300 μm, it is hard to believe that the nanocubes on the polyimide did not shift transversely (perpendicularly to the X-ray beam) for such a large amount of deformation. Therefore, I would recommend authors reword on this and give better explanation on the position shift of nanocubes.
5. With a fast growth rate (e.g. at the beginning of the reaction) and a large polyimide deformation in the ptychographic scans, did authors use any methods (e.g. position correction) in image reconstruction to improve reconstruction quality?
6. I am not sure the way of obtaining out-of-plane thickness and the aspect ratio of nanocubes in the session starting at line 206 is correct. (1) If the background is uniform, then I think the method described in the paper is fine. However, the background was “non-flat” due to “these non-static portions of the sample” as stated in line 141, the obtained phase shifts can’t be calculated into the thickness of nanocubes as they might contain non-static portions of the sample (in the solution). (2) With the consideration of a large deformation of cell thickness (from 1000 μm to below 300 μm), the conclusion deduced from Figure 4b might not be correct (line 236-238). It could be the calculation of nanocube thickness by the above method became more accurate as the background effect became less due to the decrease of the cell thickness.
7. In Figure 6a, there were few nanocubes visible in the exposed region, is it possible that they were detached from the window into solution instead of disintegration? Authors also stated that 8.98 keV photon energy had more radiation damage on the sample than 15.25 keV, why can’t we see this “disintegration” at the end of the reaction using 8.98 keV?
8. In the conclusion, it is not quite clear why the “beam damage will be further reduced while significantly increasing imaging speed and spatial resolution”. Increasing spatial resolution usually means more X-ray dose deposited on sample.
9. In supplementary note 3, there is one sentence written in German.

Reviewer #2 (Remarks to the Author):

This paper discusses a how X-ray ptychography can image the in situ formation and hollowing of nanoparticles in solution. The paper is well written and well presented. The development of new in situ methodologies from beamlines and inside electron microscopes is essential for the understanding of how nanoparticles and nanomaterials grow.

This paper is an excellent and well executed study looking at nanocubes in which 2 processes are monitored and followed, with the observation that the reduction of Cu₂O to Cu can initiate the hollowing of the nanocuboids.

The X-ray technique has improving resolution and can now reach a resolution of 66 nm which is becoming comparable with electron microscopy.

Can the authors provide any more information about the kinetics beyond.

"that they resemble the pseudo first-order kinetic"

Any rate constants etc?

Reviewer #3 (Remarks to the Author):

My expertise means I can comment for the most part on the ptychographic imaging experiment and algorithms reported in this manuscript; I am less able to provide expert input on the significance and potential impact of the application.

The manuscript describes high resolution ptychographic X-ray imaging of cuprous oxide nanocubes in a bespoke sample chamber. There are several novel aspects to the experiment and reconstruction that make it interesting: the fact that the sample is dynamic and in solution demonstrates a surprisingly high degree of robustness for a computational technique like this, and the use of multislice to image the front and back surfaces of the chamber is novel and an ideal use-case for this new method. These features added to the application detail together represent a significant body of work for which I commend the authors. It is clear from the video in the supplement that reconstructing this data must have been difficult.

The paper is nicely presented and well written and I have no issue with the technical content. The degree of noise and distortion affecting the results means I'm less sure about the strength of the analysis and conclusions, for example extrapolating sample thickness from the heavily distorted phase images would seem prone to significant error.

I would have liked to have seen more detail of the multislice reconstructions, e.g. where the depth of field becomes smaller and the separation of the front and back surfaces begins to fail, but that is perhaps not directly relevant to the results and simply a reflection of my own interests.

The suitability of the paper for Nature Comms I think relies on expert opinion on the significance of the application findings around the growth of the nanocubes and their structural properties, which I cannot provide. This is a nice paper, though, that demonstrates both the high robustness and the potential usefulness of multislice ptychography.

Reviewer #4 (Remarks to the Author):

In this manuscript, the authors demonstrate in-situ imaging of the morphology transformation

during the preparation of Cu hollow cuboids by X-ray ptychography. Based on quantitative phase analysis, the authors are able to obtain the thickness information of the crystals and reveal the hollow interiors of the final products. As the study mainly focuses on the crystals attached to the polyimide windows of the in situ reactor, the authors observe some different growth behaviors of crystals attached to the substrates, e.g., the crystals would tend to grow from only one face in the direction perpendicular to the substrate while grow from all four side faces, resulting in the formation of cuboids with an aspect ratio around 0.5. Further study indicates that the hollowing process of the crystals originates from the center of the particle-substrate interface.

X-ray ptychography, as well as quantitative phase analysis for thickness detection has already been reported by the authors in their previous work [Scientific Reports 11, 1500 (2021)]. The originality of the present work manuscript lies in the in situ observation of hollow interiors within the final crystals. Although the in situ observation of hollowing process could be potentially interesting from a fundamental perspective, showing the capability of X-ray ptychography in monitoring the formation path of hollow nanocrystals, the resolution (~ 66 nm) of the current study is far from adequate of the practical application. Moreover, the size of the selected Cu₂O cuboids is over a few hundred nanometers and cannot be called nanocubes. A number of critical flaws are detailed below, resulting in (unfortunately) a poor significance of the work presented by the authors.

1)The main flaw in this study lies in the resolution of the X-ray ptychography. Although hollow nanocrystals have attracted numerous attentions in recent years, the reported in situ imaging method is not suitable for any hollow nanocrystals due to their low resolution (~ 66 nm in this work). The selected Cu₂O cuboids were of size around ~ 500 nm, which cannot be recognized as nanocubes.

2)The mechanism for the formation of void is attributed speculatively to the course of reduction, different from the nanoscale kirkendall effect [see. Chem Mater 25, 1179–1189 (2013) and Science 304, 711-714 (2004)]. The reduction, in most cases, can only take place at the surface of the crystals. If so, how can voids form inside the crystals? Does O species diffuse outward here? Could the origin of the hollow interior be attributed to Kirkendall effect? How to reconcile these two claims? These points should be discussed.

3)In Figure 4b, the authors show the evolution of the aspect ratio for Cu₂O cuboids. For cuboids nucleated on substrates, the aspect ratio gradually increased from ~ 0.2 to ~ 0.5 along with the reaction. Why and how can cuboids with aspect ratio of ~ 0.2 form on the substrate? According to the growth theory proposed by the authors, the “correct” aspect ratio should be around 0.5 throughout the reaction period. These points should be clarified.

Collectively, the results do not provide a “deep understanding of the growth mechanism of hollow nanocrystals” due to the low resolution of in situ imaging method and part of improper and unclear explanation for the growth mechanism. These matters concern the key idea and conclusions of the paper; hence it is not recommended for the publication in Nat Commun.

We thank the reviewers for their valuable comments. We are pleased that they mostly share our enthusiasm for in situ X-ray microscopy and recognize its potential for nanomaterials synthesis. We have revised the manuscript complying with the reviewers' comments. Please find below our detailed response to their comments:

The reviewers' comments are written in black, while *our responses are written in blue and italic.*

Reviewer 1:

This manuscript by Grote et.al. presents the use of X-ray ptychography to study the growth of Cu₂O nanocubes and the formation of hollow structure during the reduction of Cu₂O to Cu. In general, this is an interesting work which shows a powerful demonstration of the utility of X-ray ptychography in in-situ characterization of materials growth & reaction.

We thank the reviewer for her/his positive evaluation of our work.

Here are some of my comments that could improve the work:

1. The use of "in 3D" in the title is misleading. Even though multi-slice ptychography was used in this paper to distinguish the entrance and exit planes that were separated by more than 280 microns, only 2D slices (e.g. the entrance window) which are the "projections of the electron density" as stated in line 165 were used for the data analysis.

We agree with the reviewer that the use of "in 3D" in the title could be misunderstood. Our intention was to reflect the 3D modelling of the growth process achieved by the quantitative analysis. We deleted the term "in 3D" from the title.

2. Authors should consider including some early papers of in-situ X-ray ptychography in the introduction. For example:

Høydalsvik et al. <https://doi.org/10.1063/1.4884598>

Fløystad et al. [https://DOI: 10.1002/adem.201400443](https://DOI:10.1002/adem.201400443)

Strelnikova et al. <https://DOI:10.1038/s41598-017-13175-9>

We thank the reviewer for drawing our attention to these important early in situ experiments. We included the work by Høydalsvik et al. (2014) and Fløystad et al. (2015) in lines 65-68 of the Introduction section.

3. In line 87-88, it is not quite clear what the "single-digit nanometer range" is?

We clarified this point by rephrasing the sentence about the thermal simulations of the reactor in lines 97-99 as follows:

"By simulating the thermal expansion of the reactor for small temperature fluctuations during operation, we find that thermal drifts of the reactor itself can be expected below 10 nm."

4. In contrast to "Irrespective of small shifts" and "position of the nanocubes is stable", obvious shifts can be found in Figure 2e between two consecutive scans acquired in 10.7 h and 11.1 h. For example, the blue and red squares are unmatched especially upper left of the scale bar.

(1) Was this shift due to the reactor's thermal drift?

We thank the reviewer for pointing out that we missed to give sufficient information on the question of position preservation. Already during the measurement, we mechanically adjusted the position of the reactor after each image acquisitions. The displacements were in the range of 0.5 to 3 μm . Simulations showed that the displacement of the reactor due to its thermal expansion was below 10 nm. Thus, thermal drifts of other components in the setup must have caused the comparably large displacements between acquisitions. The reactor was a strong source of heat placed as close as a few centimeters to the optical elements. In addition to the mechanical position correction, we aligned the images of the in situ series to each other using the SIFT method implemented in ImageJ.

The remaining position inaccuracy due to thermal drifts during the acquisition of a single image is likely to be the cause of the displacement of the particles in the lower left of Figure 2e. This can of course not be corrected for by image alignment. In this regard, we refer to our answer to the reviewer's comment no. 5.

We rephrased and included the necessary explanations in lines 147-157, as well as in the caption of Figure 2 (line 144), and in line 481 and 510-513 of the Methods section.

(2) A striking information provided in the manuscript was that the two polyimide windows were deformed during the reaction so that the cell thickness decreased from 1000 μm to below 300 μm , it is hard to believe that the nanocubes on the polyimide did not shift transversely (perpendicularly to the X-ray beam) for such a large amount of deformation. Therefore, I would recommend authors reword on this and give better explanation on the position shift of nanocubes.

This is correct. In addition to the thermal drifts described above, the two windows of the reactor also shifted transversely with respect to each other. This happened noticeably when the rate of distance shrinkage became higher at about 10 h reaction time. We performed the mechanical position correction during the measurement by observing the particles only on the entrance window. Thus, the FOV on the exit window continuously sifted by a maximum of about 15 μm during the entire time series.

For the analysis of growth and hollowing in Figures 1-5, however, it was necessary to follow the same region with nanocubes only on one of the windows.

5. With a fast growth rate (e.g. at the beginning of the reaction) and a large polyimide deformation in the ptychographic scans, did authors use any methods (e.g. position correction) in image reconstruction to improve reconstruction quality?

Yes. From iteration 250 of 1000, position refinement was performed every 100 iterations with a maximum displacement of 44 nm as described by [Schropp et al. Scientific Reports 3, 1633 (2013), DOI: [10.1038/srep01633](https://doi.org/10.1038/srep01633), see Methods section therein]. This is described in lines 503-505 of the Methods section.

In regard of our answer to the reviewer's comment no. 4, the position refinement allowed to correct for thermal drifts during the acquisition of a single image and improved the reconstruction quality significantly. We added a direct comparison of reconstructions with and without position refinement in Supplementary Figure 5. While in some parts of the image the shift of the nanocubes could be further reduced, some displacements in other parts of the image remained unimproved. However, these remaining shifts do not affect the information content or the later quantitative analysis of the images.

6. I am not sure the way of obtaining out-of-plane thickness and the aspect ratio of nanocubes in the session starting at line 206 is correct. (1) If the background is uniform, then I think the method

described in the paper is fine. However, the background was “non-flat” due to “these non-static portions of the sample” as stated in line 141, the obtained phase shifts can’t be calculated into the thickness of nanocubes as they might contain non-static portions of the sample (in the solution).

We thank the reviewer for pointing the discussion to this key aspect of the manuscript. Still, we are sure that the background fluctuations do not affect the measurement of particle thickness to any extent greater than it is indicated by the error bar in Figure 4b.

We already took into account the background fluctuations in the thickness calculation by (1) subtracting the average phase shift in the background area of the images from the phase shift of the particles (see lines 533-535 of the Methods section) and (2) by using the standard deviation of the background fluctuations in the error calculation of the aspect ratio (see Supplementary Notes 3).

Of course, the background fluctuations have an effect on the thickness calculation of a particle in each individual image, as it can be seen in Supplementary Figure 12. Still, due to the stochastic nature of these fluctuations, the result becomes more accurate by applying a moving average with a window width of four consecutive time steps, as it was done for the plot in Figure 4b.

Despite the possible error of the individual data points, the trend that those particles growing mostly on the substrate have only half the aspect ratio of the cubic particles mostly growing in solution is real and it was additionally confirmed by SEM ex-situ studies shown in Fig. 5h-i.

To clarify this point, we added the sentence in lines 253-255.

(2) With the consideration of a large deformation of cell thickness (from 1000 μm to below 300 μm), the conclusion deduced from Figure 4b might not be correct (line 236-238). It could be the calculation of nanocube thickness by the above method became more accurate as the background effect became less due to the decrease of the cell thickness.

At reaction times between 8 and 12 h, our analysis shows nanocuboids with different aspect ratios in the same images. While the aspect reading for those particles that were attached to the window from the beginning remains around 0.5 in all images, only the aspect ratio reading of the newly attached particles is different, irrespective from which image in the series it is calculated. We can thus conclude that the shrinking distance does not change the result of the aspect ratio measurement during the series, and according to that the aspect ratio reading must be independent of the window distance.

7. In Figure 6a, there were few nanocubes visible in the exposed region, is it possible that they were detached from the window into solution instead of disintegration?

In contrast to the sudden attachment of particles from one image to the next, we never observed a sudden detachment. As it can be seen from Supplementary Figure 17, the particles slowly disintegrate under illumination after their reduction to metallic copper. The reviewer is correct that small particles and fragments of particles are still visible in the exposed region in Figure 6a. This indicates that their disintegration was not yet complete at the time when the image was taken.

Authors also stated that 8.98 keV photon energy had more radiation damage on the sample than 15.25 keV, why can’t we see this “disintegration” at the end of the reaction using 8.98 keV?

We initially performed our in situ experiments at 8.98 keV and unfortunately we observed that the morphology, composition and structure of nanoparticles grown ex situ and in situ are completely different. If we looked at the images, we observe that the beam of 8.98 keV must trigger the reduction

of the copper precursor already in solution, which leads immediately to the nucleation and growth of a dense Cu particle layer. So, we do not see disintegration, but on the contrary, we see that the beam damage induces Cu particle formation. These experiments were the reason we moved to the higher X-ray energies. At 15.25 keV the morphology, composition and structure of nanoparticles grown ex situ and in situ are alike and only disintegration is observed under prolonged exposure at the same spot. Note that the imaging of ex situ sample (without solution) can be done at both energies 8.98 keV and 15.25 keV, without beam damage within 20 hs.

In general, for reaction in solution different chemical processes, such as the breaking of chemical bonds or the formation of reactive radical species, can occur at different photon energies, leading to various effective forms of radiation damage.

8. In the conclusion, it is not quite clear why the “beam damage will be further reduced while significantly increasing imaging speed and spatial resolution”. Increasing spatial resolution usually means more X-ray dose deposited on sample.

We made this statement in accordance with our observations that radiation damage was significantly reduced by increasing the photon energy. Even though the scattering power of the sample decreases with increasing photon energy, we still expect that the effect of radiation damage will be significantly less prominent at high energies. Since fourth generation synchrotron sources will provide a high coherent flux even at energies way above 15.25 keV, we believe that it will be possible to use more intense beams for higher temporal and spatial resolution while at the same time reducing radiation damage.

To avoid any misunderstanding in this regard, we rephrased the sentence in lines 424-428 as follows:

“According to our observation of reduced beam damage at the higher photon energy of 15.25 keV, we expect that beam damage will be further reduced with the emergence of fourth generation synchrotron sources providing highly coherent X-rays at even higher energies than 15.25 keV. At the same time, the high coherent flux of these new sources can significantly increase imaging speed and spatial resolution.”

9. In supplementary note 3, there is one sentence written in German.

We thank the reviewer for pointing this out. We corrected the appearance of the German sentence in Supplementary Notes 3.

Reviewer 2:

This paper discusses a how X-ray ptychography can image the in situ formation and hollowing of nanoparticles in solution. The paper is well written and well presented. The development of new in situ methodologies from beamlines and inside electron microscopes is essential for the understanding of how nanoparticles and nanomaterials grow.

This paper is an excellent and well executed study looking at nanocubes in which 2 processes are monitored and followed, with the observation that the reduction of Cu₂O to Cu can initiate the hollowing of the nanocuboids.

The X-ray technique has improving resolution and can now reach a resolution of 66 nm which is becoming comparable with electron microscopy.

We thank the reviewer for her/his positive evaluation of our work. We are pleased that the reviewer recognizes the progress that has been made in the field of X-ray microscopy and understands its importance and potential for deeper insights into nanomaterials synthesis.

Can the authors provide any more information about the kinetics beyond "that they resemble the pseudo first-order kinetic" Any rate constants etc?

We thank the reviewer for this question. Unfortunately, fluctuations in the size reading make fitting of the size evolution plot and the calculation of a rate constant prone to significant error. In addition, since the edge size of the nanocubes is not linearly related to the amount of Cu₂O formed and the range of involved cube sizes is quite wide, we concluded that the calculation of a rate constant for this reaction from the size analysis does not give a reasonable result.

Reviewer 3:

My expertise means I can comment for the most part on the ptychographic imaging experiment and algorithms reported in this manuscript; I am less able to provide expert input on the significance and potential impact of the application.

We thank the reviewer for her/his valuable input on the methodological aspects of the manuscript.

The manuscript describes high resolution ptychographic X-ray imaging of cuprous oxide nanocubes in a bespoke sample chamber. There are several novel aspects to the experiment and reconstruction that make it interesting: the fact that the sample is dynamic and in solution demonstrates a surprisingly high degree of robustness for a computational technique like this, and the use of multislice to image the front and back surfaces of the chamber is novel and an ideal use-case for this new method. These features added to the application detail together represent a significant body of work for which I commend the authors. It is clear from the video in the supplement that reconstructing this data must have been difficult.

The paper is nicely presented and well written and I have no issue with the technical content.

We thank the reviewer for her/his positive evaluation of our work. We especially appreciate that the reviewer points out the robustness of ptychographic image reconstruction which, combined with multislicing, make it an ideal tool for in situ visualization in materials science.

The degree of noise and distortion affecting the results means I'm less sure about the strength of the analysis and conclusions, for example extrapolating sample thickness from the heavily distorted phase images would seem prone to significant error.

We thank the reviewer for pointing the discussion to this key aspect of the manuscript. We are nevertheless sure that the background noise and distortions do not affect the measurement of particle thickness to any extent greater than it is indicated by the error bar in Figure 4b.

Since this comment is in essence similar to comment no. 6 of Reviewer 1, we would like to refer to our answer there.

I would have liked to have seen more detail of the multislice reconstructions, e.g. where the depth of field becomes smaller and the separation of the front and back surfaces begins to fail, but that is perhaps not directly relevant to the results and simply a reflection of my own interests.

*We thank the reviewer for posing this interesting question about the failing slice separation at low slice distance. To this end we added Supplementary Figure 8 showing an example of a close-to-failing multi-slice reconstruction for a slice separation of 400 μm . The depth of field of the experiment was 280 μm , thus a multi-slice reconstruction is technically possible. Still, we observe that particles in the entrance window slice start to become dimly visible also in the exit window slice, indicating an incomplete slice separation. At even smaller distances, a multi-slice reconstruction fails to provide any reasonable image. The methodology was described in our previous work on model samples [Kahnt et al. Scientific Reports **11**, 1 (2021), DOI: [10.1038/s41598-020-80926-6](https://doi.org/10.1038/s41598-020-80926-6)]. There, we systematically investigate the question how a slice distance in the range of the depth of field influences the lateral image resolution when reconstructed with a single- or multi-slice approach.*

We added an explanation of this effect in lines 176-179.

The suitability of the paper for Nature Comms I think relies on expert opinion on the significance of the application findings around the growth of the nanocubes and their structural properties, which I cannot provide. This is a nice paper, though, that demonstrates both the high robustness and the potential usefulness of multislice ptychography.

Reviewer 4:

In this manuscript, the authors demonstrate in-situ imaging of the morphology transformation during the preparation of Cu hollow cuboids by X-ray ptychography. Based on quantitative phase analysis, the authors are able to obtain the thickness information of the crystals and reveal the hollow interiors of the final products. As the study mainly focuses on the crystals attached to the polyimide windows of the in situ reactor, the authors observe some different growth behaviors of crystals attached to the substrates, e.g., the crystals would tend to grow from only one face in the direction perpendicular to the substrate while grow from all four side faces, resulting in the formation of cuboids with an aspect ratio around 0.5. Further study indicates that the hollowing process of the crystals originates from the center of the particle-substrate interface.

X-ray ptychography, as well as quantitative phase analysis for thickness detection has already been reported by the authors in their previous work [Scientific Reports **11**, 1500 (2021)]. The originality of the present work manuscript lies in the in situ observation of hollow interiors within the final crystals.

We thank the reviewer for making this connection with our previous work. However, we disagree with the reviewer concerning the originality of the current manuscript.

*In our previous publication [Kahnt et al. Scientific Reports **11**, 1 (2021), DOI: [10.1038/s41598-020-80926-6](https://doi.org/10.1038/s41598-020-80926-6)], we did not perform any quantitative analysis of phase images and we did not study particle thickness. The previous work is limited to studying the effect of slice distance in multi-sliced ptychography, thus the effect of reactor thickness, on the image resolution.*

Thus, in addition to demonstrating the applicability of X-ray ptychography in a heated solution, the novelty of the current manuscript includes the in situ detection of particle thickness.

Although the in situ observation of hollowing process could be potentially interesting from a fundamental perspective, showing the capability of X-ray ptychography in monitoring the formation path of hollow nanocrystals, the resolution (~ 66 nm) of the current study is far from adequate of the practical application.

We concede that the presented resolution, which we obtained after this first demonstration of in situ ptychography in a heated solution, is not yet comparable with electron microscopy. Even with new IV generation synchrotron sources and stable microscopes we will not be reaching the sub nm resolution. Here we show that instead, X-ray ptychography nicely complements and extends the application range of microscopic methods. In situ X-ray microscopy enables visual insights in many areas of materials science with are currently not accessible with in situ electron microscopy, as X-rays can penetrate thick liquid and even solid environments. Thus, the power of the method relies on this enabling character.

Worthwhile practical applications include lithium ion batteries, for which hollow composited metal oxide nanoparticles of similar size as our nanocuboids are often used as electrode materials.

*[Xia et al, Nature Communications **12**, 2973 (2021), DOI: [10.1038/s41467-021-23150-8](https://doi.org/10.1038/s41467-021-23150-8)]*

*[Seh et al, Nature Communications **4**, 1331 (2013), DOI: [10.1038/ncomms2327](https://doi.org/10.1038/ncomms2327)]*

On the other hand, void formation in this size range is a prominent failure mechanism during cycling of lithium ion batteries and may thus benefit from ptychographic operando studies.

*[Lewis et al, Nature Materials **20**, 503–510 (2021), DOI: [10.1038/s41563-020-00903-2](https://doi.org/10.1038/s41563-020-00903-2)]*

*[Kuratani et al, ACS Appl. Energy Mater. **3**, 5472–5478 (2020), DOI: [10.1021/acsaem.0c00460](https://doi.org/10.1021/acsaem.0c00460)]*

Moreover, hollow nanospheres made of noble metal and metal oxide composites are promising candidates in (photo-) catalytic energy production.

[Xiao et al, Nature Materials (2022), DOI: [10.1038/s41563-021-01183-0](https://doi.org/10.1038/s41563-021-01183-0)]

*[Sun et al, Nature Communications **3**, 1139 (2012), DOI: [10.1038/ncomms2152](https://doi.org/10.1038/ncomms2152)]*

Void formation in the sub micrometer range due to electromigration is a common failure mechanism in integrated electronic circuits currently studied only ex situ using electron microscopy. Here, operando X-ray microscopy can be helpful.

*[Jin et al, Scientific Reports **11**, 8668 (2021), DOI: [10.1038/s41598-021-88122-w](https://doi.org/10.1038/s41598-021-88122-w)]*

*[Chang et al, Scientific Reports **7**, 17950 (2017), DOI: [10.1038/s41598-017-06250-8](https://doi.org/10.1038/s41598-017-06250-8)]*

*[Ceric et al, Materials Science and Engineering R **71**, 53–86 (2011), DOI: [10.1016/j.mser.2010.09.001](https://doi.org/10.1016/j.mser.2010.09.001)]*

Also, in biomineralization and bio-inspired synthetic materials, which show for example extraordinary mechanical properties, the nanostructures often have sizes in a range accessible with our method.

*[Yang et al, Science **375**, 647–652 (2022), DOI: [10.1126/science.abj9472](https://doi.org/10.1126/science.abj9472)]*

*[Jiang et al, Science **368**, 642–648 (2020), DOI: [10.1126/science.aaz7949](https://doi.org/10.1126/science.aaz7949)]*

*[Couasnon et al, Science Advances **6**, eaaz3125 (2020), DOI: [10.1126/sciadv.aaz3125](https://doi.org/10.1126/sciadv.aaz3125)]*

*[Jang et al, Nature Materials **12**, 893-898 (2013), DOI: [10.1038/NMAT3738](https://doi.org/10.1038/NMAT3738)]*

All these applications can potentially benefit from in situ monitoring of the nanomaterials formation process to gain better control and enhanced functionality. Thus, we consider our work an important advancement with a general relevance to the various mentioned research fields.

We agree with the reviewer that the examples of hollow nanomaterials that we referenced in the Introduction (references 1-10) and in the Discussion sections (references 61, 65-66) of the original manuscript were mostly not in the size range observable with a resolution of ~66 nm. To better contextualize our work with other potential applications, we made changes to these sections.

We changed the Introduction section in lines 37-39 and we changed the outlook at the end of the Discussion section in lines 428-433 to include the above examples.

Moreover, the size of the selected Cu₂O cuboids is over a few hundred nanometers and cannot be called nanocubes.

We agree that the particles under study in this manuscript are larger than many other nanomaterials being researched nowadays. Still, we disagree with the reviewer regarding the use of the prefix “nano”. In accordance with our literature research, it is usually appropriate to use the prefix “nano” for materials with feature sizes clearly below 1 μm. The hollow nanocuboids studied in the presented manuscript have an average outer dimension of 450 nm and a central void with an average size of only 200 nm (see Supplementary Figure 14). Thus, they fall within the “nano” size regime.

We would kindly like to refer to the following examples from the literature in which materials of sizes similar to our particles are referred to as “nano”:

- *[Xia et al. Nature Communications **12**, 2973 (2021), DOI: [10.1038/s41467-021-23150-8](https://doi.org/10.1038/s41467-021-23150-8)] Figure 2: hollow Fe₃O₄@C nanospheres with a size of 420 nm*
- *[Kim et al. Small **14**, 1703432 (2018), DOI: [10.1002/smll.201703432](https://doi.org/10.1002/smll.201703432)] Figure 1c: silica nanospheres with a size of 500 nm*
- *[Shu et al. Journal of Materials Chemistry C **8**, 2913 (2020), DOI: [10.1039/c9tc05658k](https://doi.org/10.1039/c9tc05658k)] Figures 4 and 5: hollow Fe₃O₄ nanospheres with diameters of 300 to 400 nm*
- *[Sun et al. Nature Communications **3**, 1139 (2012), DOI: [10.1038/ncomms2152](https://doi.org/10.1038/ncomms2152)] Figure 3: hollow C₃N₄ nanospheres with sizes up to 300 nm*
- *[Hou et al. Applied Surface Science **437**, 92 (2018), DOI: [10.1016/j.apsusc.2017.12.157](https://doi.org/10.1016/j.apsusc.2017.12.157)] Figure 2: SiO₂ nanospheres with diameters of 300 to 500 nm*
- *[Gou et al. Nano Letters **3**, 2 (2003), DOI: [10.1021/nl0258776](https://doi.org/10.1021/nl0258776)] Figures 2 and 3: hollow Cu₂O nanocubes with an edge length of 450 nm*

A number of critical flaws are detailed below, resulting in (unfortunately) a poor significance of the work presented by the authors.

1) The main flaw in this study lies in the resolution of the X-ray ptychography. Although hollow nanocrystals have attracted numerous attentions in recent years, the reported in situ imaging method is not suitable for any hollow nanocrystals due to their low resolution (~66 nm in this work). The selected Cu₂O cuboids were of size around ~500 nm, which cannot be recognized as nanocubes.

In response to this comment, we would like to refer to our previous two answers.

2) The mechanism for the formation of void is attributed speculatively to the course of reduction, different from the nanoscale Kirkendall effect [see. Chem Mater 25, 1179–1189 (2013) and Science 304, 711-714 (2004)].

We thank the reviewer for this discussion about the cause of void formation.

Previous in situ XANES and PXRD studies on this reaction identified the reduction of Cu₂O to metallic Cu. See references 36, 39:

*[Kränzlin et al, Adv. Mater. Interfaces **2**, 1500094 (2015), DOI: [10.1002/admi.201500094](https://doi.org/10.1002/admi.201500094)]*

*[Staniuk et al. CrystEngComm, **17**, 6962–6971 (2015), DOI: [10.1039/c5ce00454c](https://doi.org/10.1039/c5ce00454c)]*

The reduction, in most cases, can only take place at the surface of the crystals. If so, how can voids form inside the crystals? Does O species diffuse outward here?

*Previous in situ FTIR and GC-MS studies showed the oxidation of benzyl alcohol to benzaldehyde during the reduction of Cu^+ cations to metallic Cu, accompanied by the release of excess H^+ ions. [see Staniuk et al. CrystEngComm, **17**, 6962–6971 (2015), DOI: [10.1039/c5ce00454c](https://doi.org/10.1039/c5ce00454c)] When Cu_2O is reduced in the solid state to Cu, there must be a net outward flow of O species caused by diffusion. It is plausible that the outward diffusing anionic O species react with H^+ at the surface to form water.*

Could the origin of the hollow interior be attributed to Kirkendall effect? How to reconcile these two claims? These points should be discussed.

Since diffusion must be involved in this solid-state reduction, we need to conclude whether diffusion is also the cause of void formation.

*The Kirkendall effect describes void formation due to different diffusion coefficients of the metallic cations and the respective counter anion during oxidation of metallic particles. The anions introduced at the surface need to diffuse inward, and the cations need to diffuse outward. Since the outward diffusion of the metal cations is faster than the inward diffusion of the anions, the result is a central void. [for diffusion rates in Cu_2O , see Figure 8b in Wang et al, Chem. Mater. **25**, 1179–118 (2013), DOI: [10.1021/cm3030928](https://doi.org/10.1021/cm3030928)]*

However, in the course of reduction of our cuprous oxide, the net diffusive flow of the two ionic species must go in opposite directions, respectively, but their diffusion coefficients do not change. Hence, different diffusion rates and thus the Kirkendall effect cannot be the cause of void formation in the course of reduction, as we observe it in our study. Additionally, in the literature we could not find any examples of void formation during reduction which would be attributed to the Kirkendall effect.

*In our study, a different phenomenon must be the cause of void formation. Since the Cu_2O cuboids are firmly bound to the polyimide substrate, we know that there is a strong interaction between the two materials. Also, strong adsorption of benzyl alcohol at the surface of Cu_2O was observed in the previous in situ FTIR study. [compare Figure S4 in Staniuk et al. CrystEngComm, **17**, 6962–6971 (2015), DOI: [10.1039/c5ce00454c](https://doi.org/10.1039/c5ce00454c)] In combination with the high surface-to-volume ratio of the flat cuboids, it is plausible that the void formation has thermodynamic reasons. It is energetically favorable to maintain the highly interacting outer surfaces at the cost of an additional internal surface.*

We thank the reviewer for pointing out that this conclusion was not easy to understand from the original manuscript. To make the argumentation more evident to the reader, we clarified the discussion in lines 299-311.

3) In Figure 4b, the authors show the evolution of the aspect ratio for Cu_2O cuboids. For cuboids nucleated on substrates, the aspect ratio gradually increased from ~ 0.2 to ~ 0.5 along with the reaction. Why and how can cuboids with aspect ratio of ~ 0.2 form on the substrate? According to the growth theory proposed by the authors, the “correct” aspect ratio should be around 0.5 throughout the reaction period. These points should be clarified.

We thank the reviewer for pointing out this discrepancy. We would like to clarify that the data in Figure 4b shows an apparent increase of the aspect ratio from about 0.3 to about 0.5 during the first hours of the reaction. Since the nanocuboids are small during these initial hours, the limited spatial resolution leads to a fading of the phase shift signal at the particle edges. Given that we average the phase shift signal of several pixels covering the central area of the nanocubes, still this fading results in an underestimation of particle thickness and aspect ratio for very small nanocuboids. This effect is however only present during the first 5 hours of reaction time.

In order to clarify this issue, we added an explanation in lines 284-288.

Collectively, the results do not provide a “**deep understanding of the growth mechanism of hollow nanocrystals**” due to the low resolution of in situ imaging method and part of improper and unclear explanation for the growth mechanism. These matters concern the key idea and conclusions of the paper; hence it is not recommended for the publication in Nat Commun.

*We would kindly like to clarify that the sentence “**deep understanding of the growth mechanism of hollow nanocrystals**” given by the reviewer in this last comment is not found in our manuscript. We agree with the reviewer that morphology is not the only level of understanding of a chemical material synthesis. Actually, we intentionally chose the copper oxide cubes synthesis because from our previous in situ PXRD/XANES/ATR studies we have had a very good understanding of chemical processes in solution at molecular and nm scale. Despite those studies, we were missing the crucial information about the morphology (shape, hollow structures, thickness of the hollow walls) and their transformation along the chemical reaction. This manuscript in a systematic and precise way closed this gap. Moreover, the methodology can be adopted to study nanomaterials from lithium ion batteries and catalysts to biomineralization as listed in detail above.*

Reviewer comments, second round review –

Reviewer #1 (Remarks to the Author):

The authors have addressed my comments for this paper. The paper has been improved after revising. I just have a small comment that the authors of Ref. 32 are incorrect. I would recommend this paper for publication.

Reviewer #2 (Remarks to the Author):

The authors have done a comprehensive job in responding to the reviewers comments.

Reviewer #3 (Remarks to the Author):

I was largely supportive of this paper in my previous review, and the authors have addressed my limited earlier concerns in this revision.

To me, the paper represents a shift for x-ray ptychography from an interesting but esoteric technique to a useful tool, addressing real-world and important scientific problems. In this sense it is timely, and it should broaden the appeal of ptychography beyond a growing but still small group of technique developers.

I must again defer to the more knowledgeable opinions of the other reviewers on the importance and technical details concerning the nanocubes themselves.

Reviewer #4 (Remarks to the Author):

I am pleased to see that the manuscript has been improved significantly by the authors since the first time I reviewed it. The in-situ X-ray ptychography technology developed in this work shows a certain potential in characterizing nanoparticles with relatively large sizes (e.g., a few hundred nanometers), which provides a good complement to electron microscopy. Considering the importance of the reported technology in the fields listed by the authors, such as lithium battery materials, this work can be published on Nat. Commun..

1) In this work, the authors selected the formation of hollow Cu nanocuboids to verify the potential application of the in-situ X-ray ptychography technology. The authors have successfully characterized the variation in thickness of the nanocuboids with a resolution of roughly 66 nm. Although the experimental data show the formation process of the hollow structure, the author's explanation for the formation mechanism of the hollow structure is not convincing. The authors claimed that the formation of the hollow interior should be ascribed to the strong surface binding of benzyl alcohol at the surface of Cu₂O. If this conclusion is true, such strong surface adsorption may further avoid the reduction of Cu₂O to metallic Cu products. Theoretically, the void formation process is not thermodynamic favorable even if the surface is passivated by strong surface capping. Therefore, the formation mechanism of hollow Cu nanocuboids should be further considered. Previously, the Kirkendall effect has been adopted for the preparation of metallic hollow nanocages (Nat. Commun. 2017, 8, 1261). In this work, it is more likely that the reduction of Cu⁺ to Cu(0) occurs on the surface of Cu₂O nanocuboids, making the surface oxygen concentration decreases. In the meantime, the concentration gradient of oxygen would lead to the outward diffusion of oxygen species and thus result in the formation of hollow interiors.

2) Another question is about the stability of Cu₂O nanocuboids under X-rays with different energies. As questioned by Reviewer 1, the authors stated that 8.98 keV photon energy had more radiation damage on the sample than 15.25 keV. A clear explanation for this weird phenomenon is required.

3) The composition and crystalline structure of the nanocuboids and hollow products should be

further confirmed by instruments like powder XRD, HRTEM, and EDX.

We thank the reviewers for approving our first revision and for pointing out the importance of in situ ptychography for materials science.

We have revised the manuscript complying with the additional comments provided by Reviewer 4. Please find below our detailed response:

The reviewers' comments are written in black, while *our responses are written in blue and italic.*

Reviewer 1:

The authors have addressed my comments for this paper. The paper has been improved after revising. I just have a small comment that the authors of Ref. 32 are incorrect. I would recommend this paper for publication.

We thank the reviewer for her/his approval of our revision, and for pointing out the wrong citation. We corrected the authors of reference 32.

Reviewer 2:

The authors have done a comprehensive job in responding to the reviewers comments.

We thank the reviewer for her/his positive evaluation of our revision.

Reviewer 3:

I was largely supportive of this paper in my previous review, and the authors have addressed my limited earlier concerns in this revision.

To me, the paper represents a shift for x-ray ptychography from an interesting but esoteric technique to a useful tool, addressing real-world and important scientific problems. In this sense it is timely, and it should broaden the appeal of ptychography beyond a growing but still small group of technique developers.

We are pleased that the reviewer shares our enthusiasm for in situ X-ray microscopy and recognizes its potential for nanomaterials synthesis.

I must again defer to the more knowledgeable opinions of the other reviewers on the importance and technical details concerning the nanocubes themselves.

Reviewer 4:

I am pleased to see that the manuscript has been improved significantly by the authors since the first time I reviewed it. The in-situ X-ray ptychography technology developed in this work shows a certain potential in characterizing nanoparticles with relatively large sizes (e.g., a few hundred nanometers), which provides a good complement to electron microscopy. Considering the importance of the reported technology in the fields listed by the authors, such as lithium battery materials, this work can be published on Nat. Commun.

We thank the reviewer for recognizing the importance of our work for the materials research and its complementarity to electron microscopy.

1) In this work, the authors selected the formation of hollow Cu nanocuboids to verify the potential application of the in-situ X-ray ptychography technology. The authors have successfully characterized the variation in thickness of the nanocuboids with a resolution of roughly 66 nm.

We thank the reviewer for this brief summary of our work.

Although the experimental data show the formation process of the hollow structure, the author's explanation for the formation mechanism of the hollow structure is not convincing. The authors claimed that the formation of the hollow interior should be ascribed to the strong surface binding of benzyl alcohol at the surface of Cu₂O. If this conclusion is true, such strong surface adsorption may further avoid the reduction of Cu₂O to metallic Cu products.

We thank the reviewer for her/his comment. However, there is no contradiction in benzyl alcohol binding to the surface and Cu₂O getting reduced, since benzyl alcohol itself is known to act as the reducing agent in this synthesis. Indeed, benzyl alcohol acts as solvent, reducing agent and stabilizing ligand at the same time. The role of benzyl alcohol and among others the strong interaction between benzyl alcohol and Cu₂O were experimentally verified in a previous in situ ATR-IR study of this synthesis.

[ref 39, Staniuk et al, CrystEngComm 17, 36 (2015), Figure S4 therein and supplementary notes 3, DOI: [10.1039/c5ce00454c](https://doi.org/10.1039/c5ce00454c)]. Furthermore, the nanocuboids interact not only with the solvent, but also with the substrate. Both interactions support the rigidity of the outer dimension of the nanocubes and prevent a shrinkage of their surface due to the mass loss upon reduction.

Theoretically, the void formation process is not thermodynamic favorable even if the surface is passivated by strong surface capping. Therefore, the formation mechanism of hollow Cu nanocuboids should be further considered. Previously, the Kirkendall effect has been adopted for the preparation of metallic hollow nanocages (Nat. Commun. 2017, 8, 1261). In this work, it is more likely that the reduction of Cu⁺ to Cu(0) occurs on the surface of Cu₂O nanocuboids, making the surface oxygen concentration decreases. In the meantime, the concentration gradient of oxygen would lead to the outward diffusion of oxygen species and thus result in the formation of hollow interiors.

We thank the reviewer for the detailed discussion of the hollowing mechanism. We modified the corresponding paragraph (lines 273-288) to additionally discuss the occurrence of the Kirkendall effect during reduction of PdP₂ nanoparticles in the reference mentioned by the reviewer. We are convinced that, among the established void formation mechanisms, a surface-protected hollowing process is the most plausible explanation for our observations and not the Kirkendall effect. The main arguments are summarized here:

- In the reference (Nat. Commun. 2017, 8, 1261) mentioned by the reviewer, the authors clearly state that for the Kirkendall effect to occur during the reduction of a compound MX to metal M, the diffusive flux of X must be higher than that of M. However, for Cu₂O at elevated temperatures, the diffusion coefficient of Cu⁺ is by several orders of magnitude higher than that of O²⁻. This holds for most diffusion pairs in metal salts: [ref. 46, Wang et al., Chem. Mater. 25 (2013), Figure 8B therein, DOI: [10.1021/cm3030928](https://doi.org/10.1021/cm3030928)]*
- Our reaction is significantly slower than the fast extraction of phosphorous in the aforementioned reference. There, due to the very high reactivity of phosphorous with O₂, phosphorous ions are quickly extracted from amorphous PdP₂ nanoparticles at 250°C. Even though the diffusion of the metal ions is usually faster, the quick extraction is stated to overcompensate this effect. In turn, a net outward mass flow is achieved according to the Kirkendall effect in this kinetically controlled*

synthesis. Figure 5 shows that in our experiment, the hollowing process takes about 3 hrs. At this slow rate, we can exclude the possibility that the extraction of oxygen at the particle surface could reverse the faster diffusion of Cu^+ . Thus, even though there is a net outward flow of O^{2-} due to the reaction taking place at the surface, the faster diffusion of Cu^+ would always inhibit void formation in a Kirkendall scenario.

- We do not see particle growth (inflation) upon hollowing, as it is usually observed in a Kirkendall process.
[ref. 47, Yin et al, Science 304, 711-714 (2004), DOI: [10.1126/science.1096566](https://doi.org/10.1126/science.1096566)]
[ref. 45, Tianou et al. Nat. Commun. 8, 1261 (2017), DOI: [10.1038/s41467-017-01258-0](https://doi.org/10.1038/s41467-017-01258-0)]
- We furthermore do not observe the formation of a yolk-shell intermediate structure as it often occurs in the Kirkendall effect, especially for particles larger than a few tens of nm.
[Nilsson et al. Nanoscale 11, 43 (2019), DOI: [10.1039/c9nr07681f](https://doi.org/10.1039/c9nr07681f)]
[ref. 46, Wang et al. Chem. Mater. 25, 1179–1189 (2013), DOI: [10.1021/cm3030928](https://doi.org/10.1021/cm3030928)]
- The Kirkendall effect is usually found in small nanoparticles. Archer and co-workers concluded in their review that for larger particles, “it is clearly inappropriate to discuss the formation mechanism based on vacancy flux and in turn the Kirkendall process.”
[Lou et al. Adv. Mater. 20, 21 (2008), page 3996, DOI: [10.1002/adma.200800854](https://doi.org/10.1002/adma.200800854)]
[ref 49, Wang et al. Chem. Rev. 116, 10983-11060 (2016), page 11015, DOI: [10.1021/acs.chemrev.5b00731](https://doi.org/10.1021/acs.chemrev.5b00731)]

2) Another question is about the stability of Cu_2O nanocuboids under X-rays with different energies. As questioned by Reviewer 1, the authors stated that 8.98 keV photon energy had more radiation damage on the sample than 15.25 keV. A clear explanation for this weird phenomenon is required.

We would kindly like to clarify that there is not more radiation damage at 8.98 keV compared to 15.25 keV, but the specific effect of the beam damage is different. “We found that the photon energy has a strong influence on the characteristic of the radiation damage.” **To make these points clearer, we extended the discussion of the possible explanations for the beam damage characteristics in lines 348-350, 353 and 359-360.**

Note that the imaging of ex situ samples (without solution) can be done at both energies 8.98 keV and 15.25 keV, without beam damage within 20 hrs. This indicates that the different beam damage effects at the two energies result from different chemical side reactions induced by the beam. At 8.98 keV, the beam causes the formation of non-cubic copper-based structures on the reactor window. At 15.25 keV though, the beam does not alter the reaction pathway and Cu_2O nanocubes form. Then, after their chemical reduction to metallic copper, the particles vanish in the beam.

In general, local energy deposition due to a partial absorption of the X-ray beam can cause changes in the reaction pathway. The energy of 8.98 keV is just above the K edge of copper, therefore a significantly larger portion of the beam is absorbed compared to 15.25 keV (the attenuation coefficient of Cu_2O is about 4 times higher at 8.98 keV than at 15.25 keV). The higher absorption at 8.98 keV is likely to trigger the reduction of the copper precursor already in solution, which means a deviation from the usual reaction pathway, leading to the nucleation and growth of a Cu particle layer. At 15.25 keV, the absorption is not strong enough to trigger the reduction of the precursor. Thus here, the reaction proceeds in its usual way until hollow metallic nanocuboids form from the solid Cu_2O particles. Metallic Cu has a higher density than Cu_2O . Therefore, even though the absorption is generally weak at 15.25 keV, it becomes stronger after the hollowing process and the transformation to metallic Cu.

Editorial Note: The figures on this page of the Peer Review File are reproduced with permission from Royal Society of Chemistry, M. Staniuk, D. Zindel, W. van Beek, O. Hirsch, N. Kränzlin, M. Niederberger and D. Koziej, *CrystEngComm*, 2015, 17, 6962 DOI: 10.1039/C5CE00454C, CC-BY 3.0

At the same time, metallic Cu is chemically less stable than Cu_2O . For these reasons, the hollow metallic nanocuboids undergo side reactions with benzyl alcohol forming soluble copper species.

3) The composition and crystalline structure of the nanocuboids and hollow products should be further confirmed by instruments like powder XRD, HRTEM, and EDX.

We thank the reviewer for this suggestion. The change from Cu_2O to metallic Cu we already verified with our previous *in situ* PXRD and XANES measurements, but these experiments lack the spatial resolution and do not provide any information about morphological changes in solution. Actually, to close this gap was the main motivation for our *in situ* X-ray ptychography studies.

In situ XANES, *in situ* ATR: ref. 39, Staniuk et al. *CrystEngComm*, 17, 6962–6971 (2015)

In situ PXRD: ref. 36, Kränzlin et al, *Adv. Mater. Interfaces* 2, 1500094 (2015)

Figure R1. In situ XANES study of the formation of Cu_2O nanocuboids from $\text{Cu}(\text{acac})_2$ and their transformation to metallic Cu. a)-c) Verification of the *in situ* data with reference spectra. d) Time-dependent composition of the reaction mixture showing the two-step process. Adapted from [ref. 39]

Moreover, we tried other methods (EDX and TEM) suggested by the reviewer, but these standard techniques were not capable of differentiating solid and hollow nanocubes and confirming their composition.

- *The polyimide window material does not allow differentiating Cu₂O from Cu with EDX, since the substrate also contains oxygen. The contribution from the substrate cannot be quantified or excluded from the EDX measurements.*
- *We could not obtain transmission images in the TEM even at 200 kV, and thus correlating shape and composition in the TEM was not possible (see Figure R3).*

Figure R3. TEM image of nanocuboids synthesized for 14 hrs and detached from the polyimide substrate. At this reaction time, the X-ray images revealed hollow nanocuboids. The image was taken with an acceleration voltage of 200 kV.

In summary, X-ray microscopy is the only method capable of visualizing the shape transformation of these substrate-bound nanocuboids due to the high penetration power of X-rays.

Reviewer comments, third round review –

Reviewer #4 (Remarks to the Author):

The authors have successfully addressed the concerns that I raised. The formation mechanism of hollow nanocubes have been carefully discussed in detail, deepening our understanding of the growth process of hollow nanocrystals. From my point of view, this work can be published in Nat. Commun. in its current form.